# ER – lysosome contacts at a pre-axonal region regulate axonal lysosome availability

Nazmiye Özkan ⬤ [1], Max Koppers ⬤ [1], Inge van Soest[1], Alexandra van Harten[1], Daphne Jurriens[1], Nalan Liv ⬤ [2], Judith Klumperman ⬤ [2], Lukas C. Kapitein ⬤ [1], Casper C. Hoogenraad ⬤ [1] & Ginny G. Farías ⬤ [1✉]

Neuronal function relies on careful coordination of organelle organization and transport. Kinesin-1 mediates transport of the endoplasmic reticulum (ER) and lysosomes into the axon and it is increasingly recognized that contacts between the ER and lysosomes influence organelle organization. However, it is unclear how organelle organization, inter-organelle communication and transport are linked and how this contributes to local organelle availability in neurons. Here, we show that somatic ER tubules are required for proper lysosome transport into the axon. Somatic ER tubule disruption causes accumulation of enlarged and less motile lysosomes at the soma. ER tubules regulate lysosome size and axonal translocation by promoting lysosome homo-fission. ER tubule – lysosome contacts often occur at a somatic pre-axonal region, where the kinesin-1-binding ER-protein P180 binds microtubules to promote kinesin-1-powered lysosome fission and subsequent axonal translocation. We propose that ER tubule – lysosome contacts at a pre-axonal region finely orchestrate axonal lysosome availability for proper neuronal function.

[1] Cell Biology, Neurobiology and Biophysics. Department of Biology, Faculty of Science, Utrecht University, Utrecht, The Netherlands. [2] Section Cell Biology, Center for Molecular Medicine, University Medical Center Utrecht,  Utrecht University, Utrecht, The Netherlands. ✉email: g.c.fariasgaldames@uu.nl

Neuronal organelle organization, functioning and transport must be carefully orchestrated to maintain neuronal architecture and function[1,2]. Microtubule (MT)-driven motor–organelle coupling ensures proper organelle transport into the two morphologically and functionally distinct structures of a neuron, the somatodendritic and axonal domains[1,3,4]. From extensive studies in non-neuronal cells, it has been increasingly recognized that organelles form contacts with each other to execute essential processes such as lipid and ion transfer, organelle division and motor transfer[5–7]. However, little is known about how organelle organization, inter-organelle communication and transport are linked and how this impacts local organelle availability in neurons.

The endoplasmic reticulum (ER) is one of the largest organelles and forms extensive contacts with various other organelles, including late endosomes (LEs)/lysosomes[8,9]. The ER is organized as perinuclear ER cisternae connected with a network of ER tubules that spread into the cell periphery of unpolarized cells[10,11]. In neurons, ER tubules are distributed along the somatodendritic and axonal domains, while ER cisternae are restricted to the somatodendritic domain[12,13]. The shape of the ER is maintained by ER-shaping proteins such as reticulons (RTNs) and DP1, which induce the curvature of tubules, and CLIMP63, which generates flattened ER cisternae[10,14,15]. Recent evidence has revealed that the ER is highly dynamic, undergoing fast remodeling in the order of seconds[9,16]. Although contacts between ER tubules and LEs/lysosomes have been visualized in both unpolarized cells and in neurons from brain tissue[8,12], it is less clear how ER remodeling regulates these organelle interactions.

It is well known that the ER and LEs/lysosomes form contacts at membrane contact sites, where small molecules and lipids can be transported reciprocally[7,17]. To maintain a steady-state number and size and correct positioning of LEs/lysosomes, essential for cellular homeostasis, they undergo series of fusion, fission, and motor-based transport events[18]. LE/lysosome fission and motor loading onto LEs/lysosomes have both been reported to occur in contact with the ER in unpolarized cells. These contacts often associate to MTs[8,9,19,20]. Yet, it remains unclear how local ER organization regulates LE/lysosome size and how this is linked to motor transfer and MT interaction at contact sites in neurons.

Proper organization and transport of ER tubules and LEs/lysosomes are crucial for neuronal development and function. ER tubules and LEs/lysosomes are translocated from the soma into the axon by the kinesin-1 motor[13,21]. Local availability of ER tubules instructs axon formation and regulates axonal synaptic vesicle cycling[13,22] and active transport of LEs/lysosomes into the axon is required for proper clearance of faulty proteins and organelles located far away from the cell soma[21,23]. Interestingly, mutations in genes encoding ER-shaping proteins cause the neurodegenerative disease hereditary spastic paraplegia, in which aberrant LE/lysosomes have been observed[10,17,24]. Therefore, it is important to understand how the organization of the ER and inter-organelle communication contribute to LE/lysosome organization and local availability in neurons.

Here, we show that ER shape regulates local LE/lysosome availability in neurons, in which somatic ER tubules promote lysosome translocation into the axon. Disruption of somatic ER tubules causes accumulation of enlarged and less motile mature lysosomes in the soma, mainly due to impaired lysosome homofission. We find that ER tubule – LE/lysosome contacts are enriched in a pre-axonal region. The MT- and kinesin-1-binding ER protein P180 is enriched and co-distributed with kinesin-1-decorated axonal MT tracks in the same pre-axonal region, where it promotes LE/lysosome motility, fission and axonal translocation. Together, our results support a model in which ER – LE/

lysosome contacts at a pre-axonal region finely orchestrate axonal LE/lysosome availability.

## Results

**ER shape regulates lysosome availability in the axon.** To study the role of the ER in neuronal LE/lysosome organization, we first investigated whether ER shape regulates LE/lysosome distribution in primary cultures of rat hippocampal neurons. ER tubules are generated by two main ER tubule-shaping proteins, RTN4 and DP1, while flattened ER cisternae are maintained by CLIMP63. The abundance of these ER-shaping proteins regulates the conversion between cisternae and tubules in cell lines and in neurons[13–15] and we therefore set out to modulate ER morphology through the knockdown of RTN4 and DP1. We confirmed the effect of knockdown on ER morphology using Ten-fold Robust Expansion microscopy (TREx)[25], which revealed a reduction in ER tubules and an increased appearance of large sheet-like structures (see Supplementary Fig. 1 and Supplementary Movie 1).

We next knocked down both RTN4 and DP1, or CLIMP63 and analyzed the distribution of GFP-tagged LAMP1, a marker for LEs and lysosomes (henceforth referred to as immature and mature lysosomes, respectively, or just lysosomes), in neurons at day-in vitro 7 (DIV7). In the control (empty pSuper vector) condition, LAMP1-positive lysosomes were abundant in the soma and evenly distributed along dendrites and the axon (Fig. 1a), as previously reported[21]. Knockdown of RTN4 plus DP1 caused a drastic reduction of LAMP1-positive lysosomes along the axon but not in dendrites, whereas CLIMP63 knockdown increased their axonal distribution (Fig. 1a; Supplementary Fig. 2a). Quantification of the polarity index (PI = [intensity dendrite −intensity axon]/[intensity dendrite + intensity axon], in which PI = 0, unpolarized; PI > 0, dendritic; P < 0, axonal distribution)[26], confirmed the unpolarized distribution of lysosomes in the control condition (PI: 0.04). Removal of ER tubule-shaping proteins disrupted axonal lysosome distribution (PI: 0.4) whereas CLIMP63 knockdown neurons showed an increased axonal lysosome distribution (PI: −0.5) (Fig. 1b). Similar results were observed using endogenously labeled LAMTOR4, another marker for lysosomes, in which ER tubule disruption caused an impaired LAMTOR4 distribution along the axon (Fig. 1c, d). Importantly, this impaired lysosome distribution is not due to cell death as cell viability was not affected (Supplementary Fig. 2b). In addition, the distribution of other organelles such as mitochondria, somatodendritic Rab11-positive recycling endosomes and axonal Rab3-positive carriers, were not altered after ER tubule disruption (Supplementary Fig. 2c, e–g, i, j), which shows that there is no general defect in organelle transport. Reduced distribution of lysosomes along the axon could be explained by an increased retrograde transport of lysosomes from the axon into the soma or an impaired translocation of lysosomes from the soma into the axon. To study this, LAMP1 dynamics was analyzed by live-cell imaging in a 30-μm-length segment of the proximal axon during a period of 300 s (Fig. 1e, f). In control neurons, an average of 32 out of 37 LAMP1-positive lysosomes per neuron were motile at the proximal axon, from which 17 transported anterogradely into the axon tip and 15 transported retrogradely to the cell soma (Fig. 1f; Supplementary Movie 3). Knockdown of RTN4 plus DP1 caused a reduction of 57.8% in the total number of lysosomes distributed along the proximal axon, decreasing the number of events for both antero- and retrograde LAMP1 movement, while the stationary pool remained unaffected (Fig. 1f; Supplementary Movie 3). These results suggest that ER tubules play a critical role in regulating lysosome translocation from the soma into the axon.

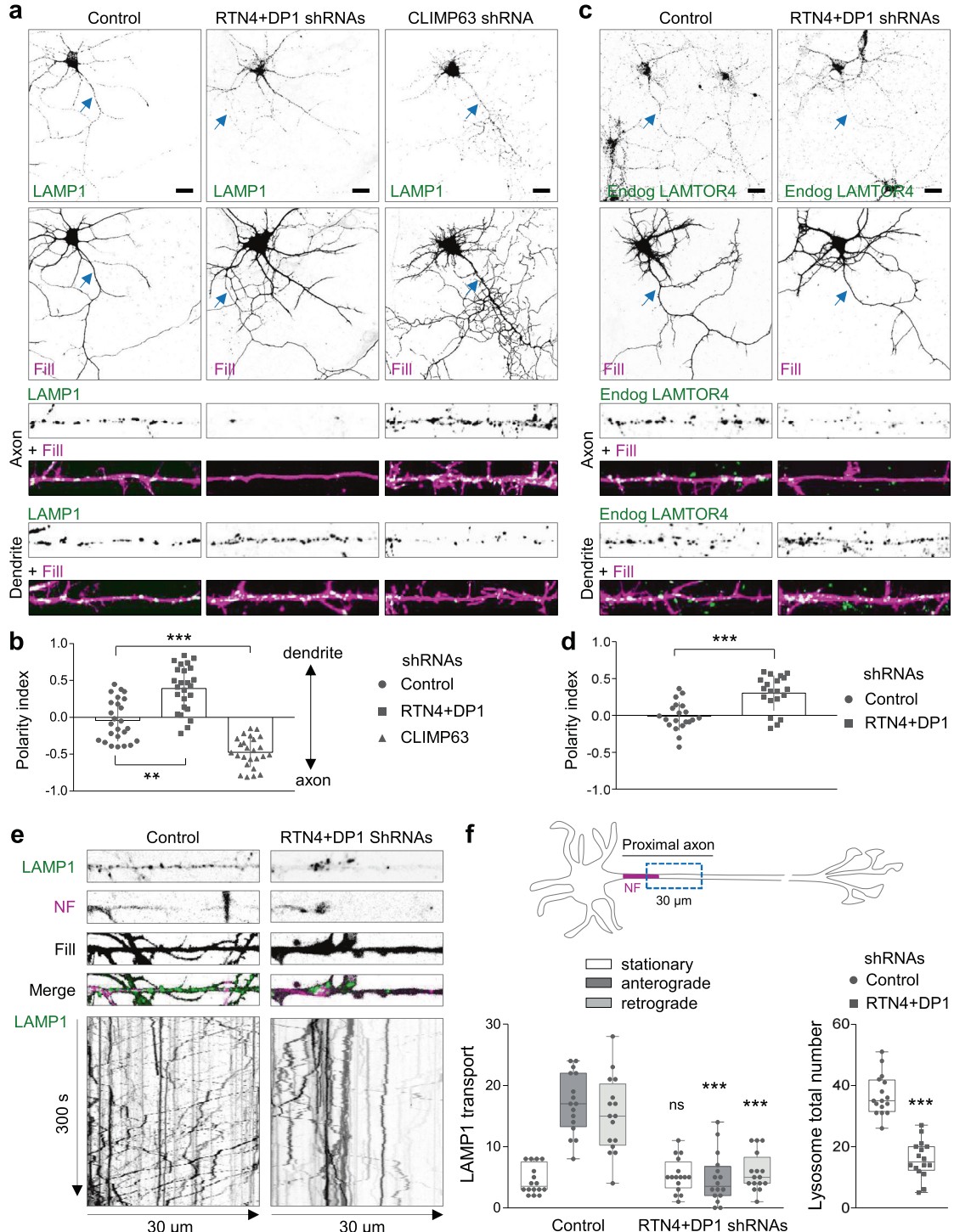

**Somatic, but not axonal, ER tubules promote lysosome translocation into the axon**. Since most of the axonal ER corresponds to ER tubules and axonal distribution of ER is impaired under RTN4 plus DP1 knockdown[12,13] (Supplementary Movie 2), we wondered whether the contacts between ER tubules and lysosomes locally regulate lysosome distribution and dynamics (Fig. 2a). To study this, we used a heterodimerization system to control ER tubule positioning. We induced a sustained retention of ER tubules in the somatodendritic domain, by triggering the binding of KIFC1 (a minus-end driven motor) to ER tubules by fusing a Streptavidin (Strep) sequence to KIFC1 and a SBP to

GFP-tagged RTN4A (Fig. 2b). This strategy allows local axonal depletion of ER membranes, as previously confirmed by the absence of several other ER markers in the axon[13]. Lysosome distribution in neurons was analyzed after 24–48 h of co-expression of LAMP1-RFP and GFP-SBP-RTN4 in the presence or absence of Strep-KIFC1. In the control condition, LAMP1 and SBP-RTN4 were co-distributed along the entire neuron, in both the somatodendritic and axonal domains (Fig. 2a). In the presence of Strep-KIFC1, axonal ER tubules containing SBP-RTN4 were pulled from the axon into the somatodendritic domain, while lysosomes were still distributed along the axon (Fig. 2b).

**Fig. 1 ER morphology controls lysosome translocation into the axon. a, b** Representative images of DIV7 hippocampal neurons co-transfected at DIV3 with LAMP1-GFP (green), a mCherry fill (magenta) and a control pSuper plasmid or a pSuper plasmid containing a shRNA sequence targeting RTN4 plus DP1, or CLIMP63, in (**a**). Higher magnification of 40-μm straightened axon (top) or dendrite (bottom) segments. Quantification of LAMP1 polarity indices in (**b**) (n = 25 neurons per condition). See also Supplementary Fig. 2a–c, e–g, i, j. **c, d** Representative images of DIV7 neurons that were transfected with a mCherry fill (magenta) and a control pSuper plasmid or with shRNAs targeting RTN4 plus DP1 and stained with a LAMTOR4 antibody (green) in (**c**). Higher magnification of 40-μm straightened axon (top) or dendrite (bottom) segments. Polarity indices for LAMTOR4 in (**d**) (n = 20 neurons per condition). **e** Representative still images (top) and kymographs (bottom) from a 30-μm-segment of straightened proximal axons of live neurons co-transfected with LAMP1-GFP (green) and fill (gray) together with control pSuper or shRNAs targeting RTN4 plus DP1, and labeled for the axon initial segment (AIS) marker Neurofascin (NF; magenta) prior to imaging for 300 s. **f** Quantification of LAMP1-positive lysosome movement and total number of lysosomes from conditions as in (**e**); n = 16 per condition. Schematic representation of a neuron indicating the axon initial segment marker NF in magenta and the axonal region (blue dotted box) selected for quantification (top), average number of stationary, anterograde and retrograde pools (bottom, left) and average number of total lysosomes (bottom, right). Blue arrows point to the proximal axon and scale bars represent 20 μm in (**a**) and (**c**). Individual data points each represent a neuron in (**b**), (**d**) and (**f**). Data are presented as mean values ± SD in (**b**) and (**d**). Boxplots show 25/75-percentiles, the median, and whiskers represent min to max values in (**f**) (ns—not significant, **p < 0.01 and ***p < 0.001 comparing conditions to control (Kruskal–Wallis test followed by a Dunn's multiple comparison test) in (**b**) and (one way ANOVA followed by a Sidak's multiple comparison test) in (**f** left), and (two-sided unpaired t-test) in (**d**) and (**f**, right). Source data and exact p values are provided as a Source Data file. See also Supplementary Movies 1–3.

We then analyzed whether axonal ER tubule removal affected the dynamics of lysosomes along the axon. Co-expression of SBP-RTN4 and Strep-KIFC1, did not cause a reduction in axonal LAMP1 motility. Antero- and retrograde movement as well as the stationary pool of LAMP1-positive lysosomes were similar to control neurons expressing only SBP-RTN4 (Fig. 2d, e, Supplementary Movie 4). These results indicate that axonally distributed ER tubules do not contribute to the availability and dynamics of lysosomes along the axon. To further examine the role of somatic ER tubules, we fused a Strep sequence to the axonal plus-end-driven kinesin-1 motor KIF5A (Fig. 2c). Co-expression of SBP-RTN4A and KIF5A-Strep induced axonal transport of somatic ER tubules, their accumulation in the distal axon and concomitant reduction in the somatodendritic domain (Fig. 2c). In these neurons, LAMP1-positive lysosomes were distributed in the somatodendritic domain but drastically reduced along the axon (Fig. 2c). Live-cell imaging showed that the total number of lysosomes along the proximal axon was impaired, and the bidirectional movement of lysosomes was drastically reduced compared to control neurons, while the stationary pool remained unaffected (Fig. 2d–f; Supplementary Movie 4). Cell viability and axonal mitochondrial distribution were not affected after somatic ER tubule redistribution into the axon (Supplementary Fig. 2b, d, h). These results indicate a role for somatic ER tubules in promoting lysosome translocation into the axon.

**Local ER tubule disruption causes enlarged and less motile mature lysosomes in the soma.** To determine how disruption of ER tubules impairs lysosome translocation into the axon, we analyzed the organization of lysosomes in the soma. ER tubule disruption caused a striking enlargement of LAMP1- or LAMTOR4-positive lysosomes in the soma but not in dendrites compared to control neurons (Fig. 3a, b; Supplementary Fig. 2a). We found that somatic enlarged lysosomes were often less motile (Supplementary Movie 5; Supplementary Fig. 3a). Similarly, somatic ER tubule redistribution into the axon induced an enlargement of lysosomes in the soma (Fig. 3c; Supplementary Fig. 3c–f). These enlarged lysosomes were also less dynamic and unable to translocate into the axon and were retained at a region preceding the axon initial segment (Supplementary Fig. 3b; Supplementary Movie 6). Redistribution of ER tubules into the soma did not significantly alter lysosome size or motility in the soma (Supplementary Fig. 3b–f, Supplementary Movie 6).

Next, we analyzed whether disruption of ER tubules alters the maturation state of these enlarged lysosomes, by analyzing the presence of active cathepsins in LAMP1-positive lysosomes (mature lysosomes). We tested two probes in live neurons;

Magic-Red, which becomes fluorescent after cathepsin-B breaks down the substrate, and SirLyso, which labels active cathepsin-D[27]. We found that ER tubule disruption and somatic ER tubule redistribution into the axon did not affect lysosome activity, as most of the less motile enlarged lysosomes contained active cathepsins, and this lysosome activity was often observed compartmentalized within their luminal domain (Fig. 3d; Supplementary Fig. 3g–i; Supplementary Movie 7). Cumulative somatic intensity for cathepsin-D, as well as protein levels for several lysosomal proteins were not affected under ER tubule disruption (Supplementary Fig 3j, k). We quantified the total number of lysosomes and mature lysosomes per soma in live neurons. The total number of LAMP1-positive lysosomes was reduced to 47.8% after ER tubule knockdown compared to control neurons, from which the mature lysosome population (LAMP1/SirLyso positive) was reduced to 32.8% (Fig. 3e). The proportion of mature lysosomes to all LAMP1-positive lysosomes was not significantly reduced after disruption of ER tubules (Fig. 3f). Quantification of the number of lysosomes by size revealed an average of 9.5 mature lysosomes per soma with a diameter bigger than 1 μm (considered as enlarged)[28] after ER tubule knockdown compared to an average of only 0.95 large mature lysosomes per soma in control neurons (Fig. 3g). The percentage of large lysosomes relative to all LAMP1-positive lysosomes, revealed that around 1.2% of mature lysosomes were larger than 1 μm in control neurons, while this number increased to 28.7% after ER tubule disruption (Fig. 3h). The average diameter of the largest mature lysosome per soma was almost doubled compared to control neurons (Fig. 3i).

To reveal the ultrastructural morphology of the enlarged LAMP1-positive lysosomes we performed correlative light electron microscopy (CLEM), by which we selected a cluster of LAMP1 and SirLyso-positive organelles for FIB.SEM (focused ion beam scanning electron microscopy) imaging[29] (Fig. 3j–o). FIB.SEM analysis showed clusters of enlarged and globular lysosomes under ER tubule disruption, which differs from the control condition (Fig. 3m–o; Supplementary Fig. 3l). The content of the aberrant lysosomes was a heterogenous mix of dense, degraded material and accumulations of intraluminal vesicles (Fig. 3l–o). The compartmentalized fluorescent SirLyso signal corresponded to areas with intraluminal vesicles, indicating that these membranes are subject to lysosomal degradation. The lysosomes showed many interaction sites with each other which extended over considerable distances, but they clearly remained separate entities. These data show that disruption of ER tubules leads to the accumulation of a collection of enlarged, globular, and enzymatically active lysosomes.

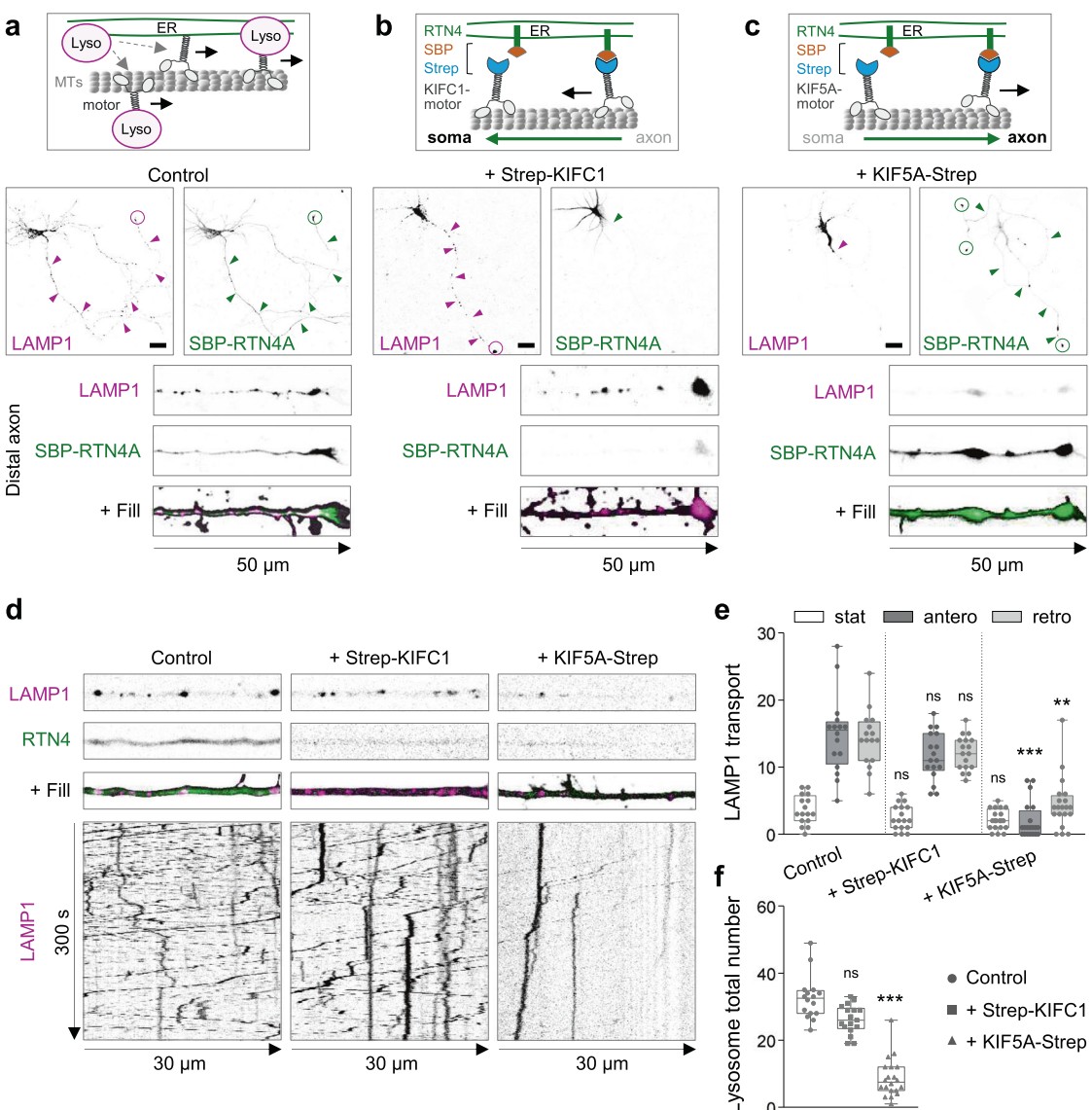

**Fig. 2 Somatic ER tubules control lysosome translocation into the axon. a** Schematic model for motor-driven lysosome transport regulated by ER tubules (top). Representative images of LAMP1-RFP (magenta) and GFP-SBP-RTN4 (green) distribution in a control DIV6 neuron co-transfected with fill (middle), and higher magnification of distal axon (bottom). **b** Schematic representation of Streptavidin (Strep)-SBP heterodimerization system using SBP-RTN4 and Strep-KIFC1 for MT-dependent minus-end ER tubule transport and its persistent somatic retention (top). Representative images of LAMP1-RFP (magenta) and GFP-SBP-RTN4 (green) distribution in a neuron co-transfected with Strep-KIFC1 and fill (middle) and higher magnification of distal axon (bottom). **c** Schematic representation of Strep-SBP system using SBP-RTN4 and KIF5A-Strep for MT-dependent anterograde transport of ER tubules and its persistent distribution in distal axons (top). Representative images of LAMP1-RFP (magenta) and GFP-SBP-RTN4 (green) distribution in a neuron co-transfected with KIF5A-Strep and fill (middle) and higher magnification of distal axon (bottom). **d**, **f** Representative still images (top) and kymographs (bottom) from a proximal axon of live neurons co-transfected with LAMP1-RFP (magenta) and SBP-RTN4 (green), in absence of a motor protein (control; $n = 16$), or with Strep-KIFC1 ($n = 17$) or KIF5A-Strep ($n = 20$) (from left to right) in **d**. Quantification of stationary, anterograde and retrograde movement of lysosomes from conditions in (**d**), in (**e**). Quantification of average total number of lysosomes from conditions in (**d**), in (**f**). See also Supplementary Movie 4. Magenta and green arrows point to the abundance of LAMP1 and SBP-RTN4 along the axon and dashed circles point to their accumulation at axon tips. Scale bars represent 20 μm in (**a**–**c**). Images are representative of three independent experiments in (**a**–**c**). Boxplot shows 25/75-percentiles, the median, and the individual datapoints each represent a neuron. Whiskers represent min to max values; ns—not significant, ***$p < 0.001$ and **$p < 0.01$ comparing conditions to control (Kruskal–Wallis test followed by a Dunn's multiple comparison test) in (**e**) and (**f**). Source data and exact $p$ values are provided as a Source Data file. See also Supplementary Fig. 2b, d, h.

Altogether, these results show that somatic ER tubules control axonal lysosome availability by regulating somatic lysosome size and motility but not activity.

**ER tubules regulate lysosome homo-fission.** Lysosome size is carefully controlled by balancing hetero-fusion or homo-fusion and fission events[18]. The enlarged LAMP1-positive structures we

observed could therefore be caused by increased fusion and/or reduced fission. We first analyzed whether enlarged lysosomes were fused with other components of the endo-lysosomal system, including early endosomes, recycling endosomes and autophagosomes, labeled by endogenous EEA1, GFP-tagged Rab11 and endogenous p62, respectively. The enlarged LAMP1- or LAMTOR4-positive lysosomes caused by ER tubule disruption

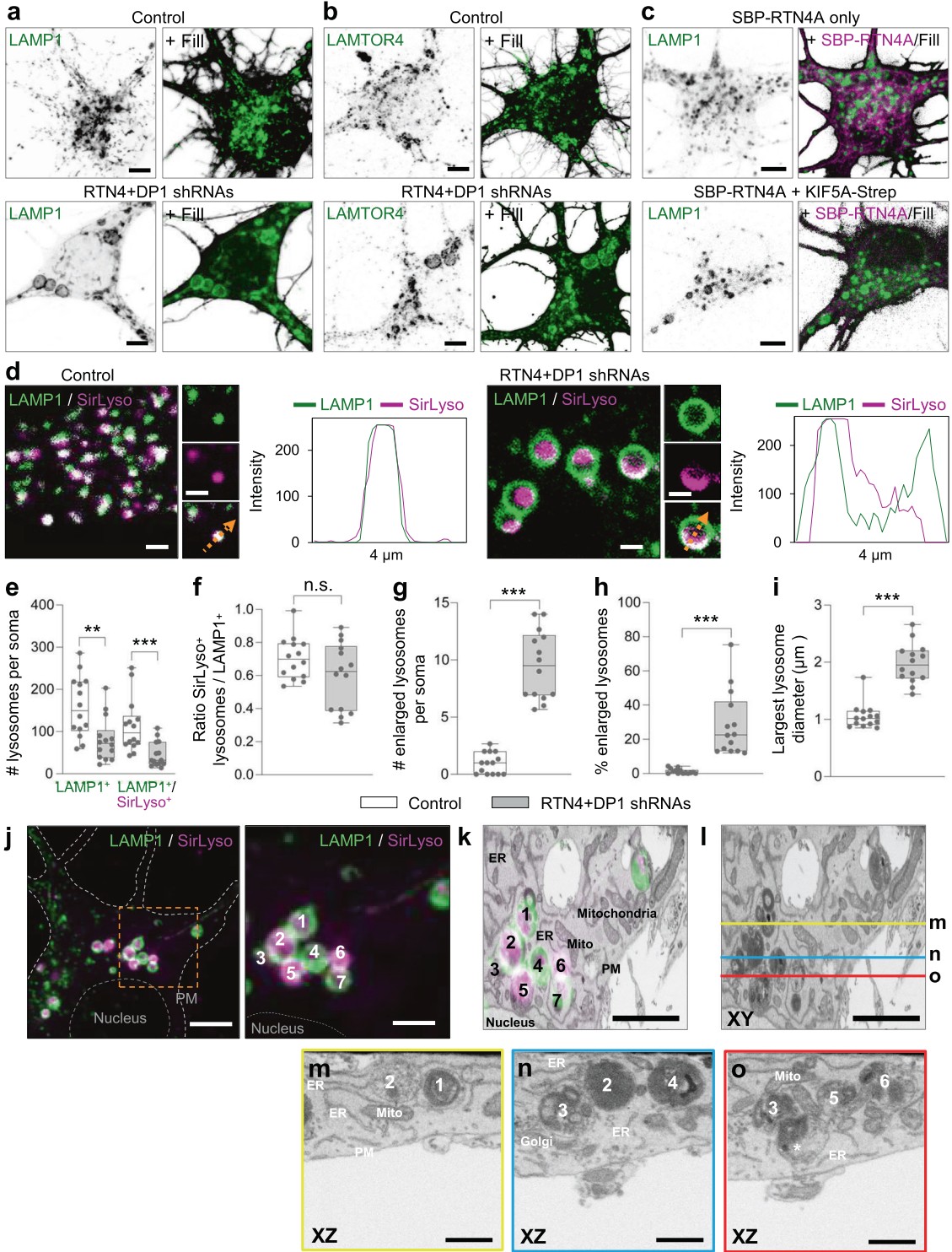

were not particularly enriched for EEA1, Rab11 or p62 markers (Supplementary Fig. 4a–c). This suggests that the enlargement of lysosomes is not induced by a mechanism that involves an increase in hetero-fusion or reduction in hetero-fission between lysosomes and early endosomes, recycling endosomes or autophagosomes.

Then, we examined whether an increase in homo-fusion and/or a reduction in homo-fission events could underlie our observed enlarged lysosomes. To determine whether homo-fusion and homo-fission were altered by ER tubule disruption, neurons expressing LAMP1-GFP were labeled with SirLyso and imaged in

the soma for a period of 300 s (Supplementary Movie 7). We focused on fusion and fission events occurring with mature lysosomes since these were more affected by ER tubule disruption. In control neurons, we observed fusion events between mature lysosomes positive for both LAMP1-GFP and SirLyso with immature lysosomes positive only for LAMP1-GFP, as well as fusion between mature lysosomes (Fig. 4a, b; Supplementary Movie 8). Fission events were also often observed, including budding of an immature or mature lysosome from a spherical mature lysosome, as well as budding from the tubular domain of a tubular-shaped mature lysosome (Fig. 4c–e; Supplementary

**Fig. 3 ER tubule disruption causes enlargement and reduced motility of mature lysosomes. a, b** Representative images of lysosomes distributed in the soma of DIV7 neurons transfected at DIV3 with fill and a control pSuper plasmid (top) or pSuper plasmids containing shRNAs targeting RTN4 plus DP1 (bottom) together with LAMP1-GFP (**a**) or stained for endogenous LAMTOR4 (**b**), green in merges. See also Supplementary Fig. 3a and Supplementary Movie 5. Images are representative of three independent experiments. **c** Representative images of lysosomes in the soma of DIV6 neurons co-transfected at DIV5 with LAMP1 (green) and fill, together with only SBP-RTN4 (magenta) as a control (top) or SBP-RTN4 plus KIF5A-Strep (bottom) to pull ER tubules into the axon. See also Supplementary Fig. 3b–f and Supplementary Movie 6. Images are representative of three independent experiments. **d** Representative still images of the soma of DIV7 neurons transfected as in (**a**) and labeled live for active cathepsin-D (magenta) with SirLyso. LAMP1 in green. The size of LAMP1-positive lysosomes and luminal distribution of cathepsin content is shown. Intensity profile line on the right of magnified image of a lysosome for each condition. See also Supplementary Fig. 3g–k and Supplementary Movie 7. **e–i** Parameters indicated in each graph were quantified from the soma of neurons transfected and labeled as in (**d**). pSuper control ($n = 14$), white bars; RTN4 plus DP1 knockdown ($n = 14$), gray bars. Boxplots show 25/75-percentiles, the median, and individual datapoints each represent a neuron. Whiskers represent min to max values; ns—not significant, ***$p < 0.001$ and **$p < 0.01$ comparing conditions to control (two-sided Mann–Whitney $U$) in (**e–i**). **j–o** Correlative light electron microscopy (CLEM) of enlarged lysosomes. (**j**) FM image of a fixed neuron, knockdown for RTN4 and DP1 and expressing LAMP1-GFP. SirLyso indicates hydrolase active lysosomes. Nucleus and plasma membrane (PM) are indicated with dashed lines. A cluster of enlarged lysosomes was selected (orange rectangle representing the ROI) for 3D-EM analyses. The right panel shows an enlargement of the ROI with seven marked lysosomes. **k** Reconstructed FIB. SEM slice of ROI in same (XY) orientation as FM and with overlay of FM signal. **l** Same EM image as in (**k**) marked with yellow, blue, and red lines that correspond to the orthogonal images shown below. **m** XZ plane image corresponding to the yellow line in (**l**) and showing cross sections of lysosomes #1 and #2. Lysosome #2 shows many intraluminal vesicles in this plane corresponding to the SirLyso signal in the FM image. **n** XZ plane image corresponding to the blue line in (**l**) and showing cross sections of lysosomes #2 and #4. In contrast to (**m**), lysosome #2 contains dense degraded material in this plane, showing the compartmentalized content of these enlarged lysosomes. Lysosome #3 and #4 are closely interacting. **o** XZ plane image corresponding to the red line in **l** showing lysosomes #3, #5 and #6. An additional lysosome (*) tightly in contact with lysosome #3 is seen in EM but not visible in the FM image. Many interactions between lysosomes were observed, but they remained separate entities. PM, Golgi, ER and mitochondria present in the EM ROI, are indicated in (**k**), (**m–o**). CLEM images are representative of three correlated samples. Scale bars represent 5 μm in (**a–c**), (**j**), 2 μm in (**d**), (**j**, right panel), (**k**) and (**l**), and 1 μm in (**m–o**). Source data and exact $p$ values are provided as a Source Data file. See also Supplementary Fig. 3l.

Movie 8). In neurons with disrupted ER tubules, fusion events between mature and immature lysosomes and between mature lysosomes were also observed (Fig. 4f, g; Supplementary Movie 8). ER tubule disruption clearly affected lysosome fission. Enlarged lysosomes often failed in the termination of the budding process to generate a mature or immature lysosome from a parent mature lysosome (Fig. 4h, i; Supplementary Movie 8). In addition, enlarged lysosomes generated instable tubules undergoing elongation followed by retraction after unsuccessful tubule fission (Fig. 4j; Supplementary Movie 8). Quantification of the number of fission events per soma and per lysosome showed that ER tubule disruption indeed caused a drastic reduction of lysosome fission events (to 17.4% and 42.67% compared to control neurons, respectively; Fig. 4k, l). We unexpectedly found that ER tubule disruption also caused a reduction in fusion events (Fig. 4m, n). Previous work in cell lines has shown that ER – endosome contacts contribute mainly to endosome fission but not fusion[8,19,30]. Our results suggest a decreased capacity of already enlarged lysosomes to fuse with other lysosomes. The number of fusion and fission events per soma were similar in control neurons with a fusion/fission ratio of 1.10, while ER tubule disruption caused an increase in this ratio to 1.44 (Fig. 4o), indicating a larger defect in lysosome fission. Impaired lysosome fission did not cause any evident lysosome stress response[31], as one of mTORC1 signaling substrates remained cytosolic, and autophagy was not increased (Supplementary Fig. 4d–h). Together, these results suggest that somatic ER tubule – lysosome contacts control lysosome fission to regulate lysosome size and translocation in neurons.

**ER tubule – lysosome contacts occur at the soma and are enriched at a pre-axonal region.** We wanted to confirm that ER tubules regulate lysosome size and axonal translocation via a direct local contact between these two organelles in the soma. To visualize the distribution of ER – lysosome contacts in neurons, we utilized the proximity-based split-APEX-labeling assay[32]. In this assay, a split version of APEX2, an engineered peroxidase which covalently tags proximal endogenous proteins with biotin in living cells, is used. Two inactive fragments, AP and EX, can

only reconstitute driven by a molecular interaction, resulting in biotinylation of a contact site[32] (Fig. 5a). We tagged protrudin, an ER tubule protein enriched in contact sites[20], with an AP module and a V5-tag, and the LE/lysosome adaptor Rab7 with an EX module. Neurons expressing the split-APEX system showed a clear co-distribution of AP-protrudin and the endogenously labeled LE/lysosome marker LAMTOR4 (Fig. 5b). The biotinylation around these two organelles, detected by fluorescently labeled Strep, indicated their co-distribution corresponds to a true contact (Fig. 5b, d). The Strep signal was specific, as only the incubation with hydrogen peroxide, which catalyzes the proximity labeling reaction, produced biotinylation (Fig. 5c). Expression of EX-Rab7-T22N, unable to bind to the lysosomal membrane[33], did not cause an enrichment of Strep signal around the ER, which suggest that interactions occur mainly when the labeled lysosomal protein is associated to the lysosomal membrane (Fig. 5d).

Importantly, we observed that ER–lysosome contacts formed mainly in the soma and they were particularly enriched in a pre-axonal region (Fig. 5e, f, i, j). ER tubule disruption caused a dramatic reduction in Strep intensity in the soma and in the pre-axonal region, compared to the control condition (Fig. 5c, g, h, k). This experiment also confirmed that these contacts occur mainly between ER tubules and lysosomes, as ER tubule disruption caused the redistribution of protrudin within more flattened ER cisternae that are excluded from the pre-axonal zone (Fig. 5g; Supplementary Movie 2). Enlarged lysosomes accumulated at the pre-axonal zone and were prevented from entering the proximal axon (Fig. 5h). Thus, ER tubules and lysosomes form contacts in the soma which are enriched in a pre-axonal region, supporting a direct role for ER tubules in lysosome fission and subsequent translocation into the axon.

**P180, a kinesin-1-binding protein enriched in ER tubules at the pre-axonal region, promotes lysosome translocation into the axon.** The kinesin-1 motor has been shown to preferentially bind axonal MTs in the soma in a region preceding the axon initial segment to promote organelle translocation into the axon[13,21,34]. Besides its role in organelle transport, kinesin-1 has

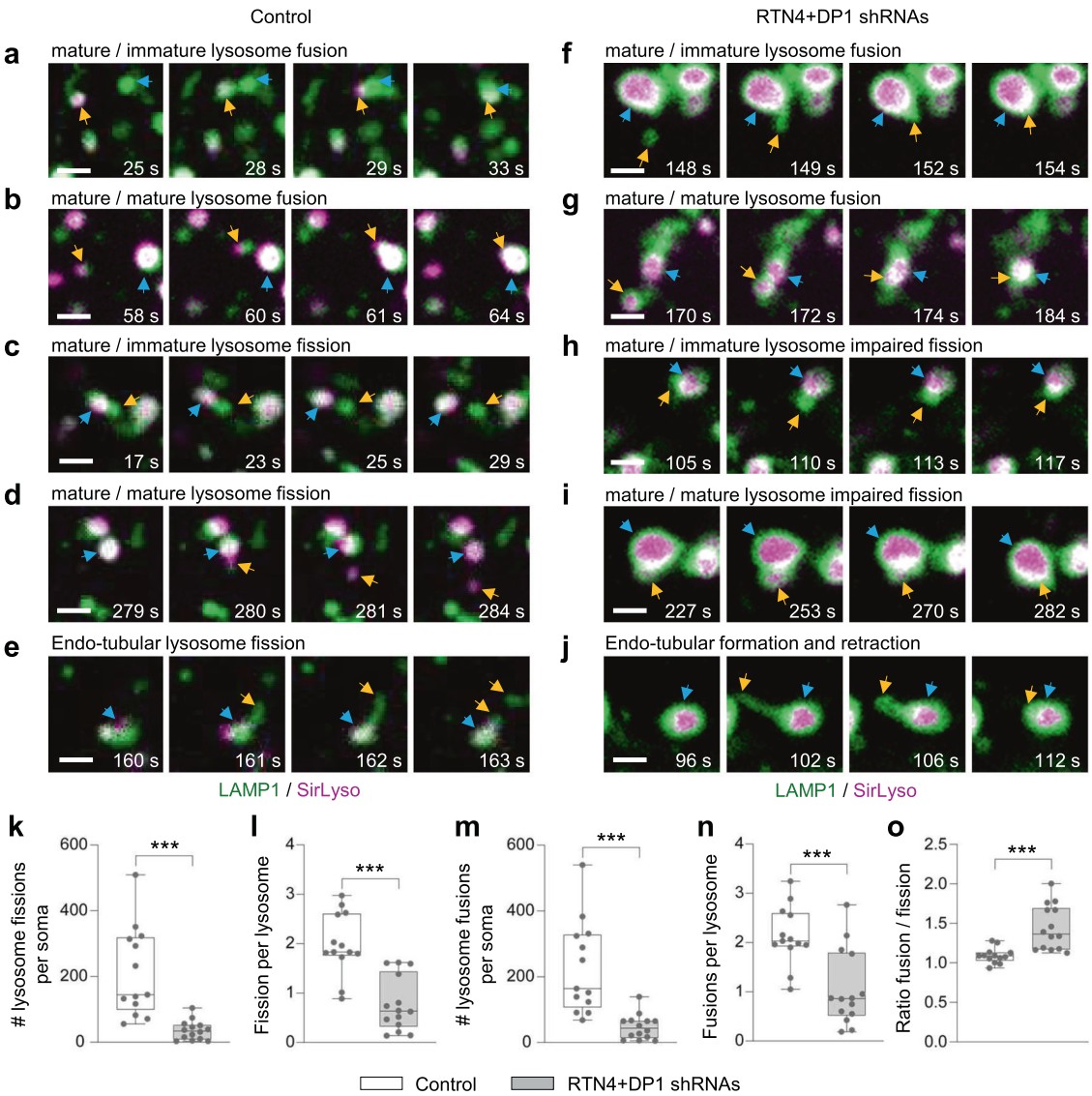

**Fig. 4 Enlarged lysosomes are caused by an imbalance in homo-fission. a–j** Representative still images of fusion and fission events from DIV7 neurons co-transfected with control pSuper (**a–e**) or shRNAs targeting RTN4 plus DP1 (**f–j**) together with LAMP1-GFP (green) and labeled with SirLyso (magenta) prior imaging for 300 s every 1 s. Fusion between lysosomes in (**a**, **b**, **f**, **g**); Fission in (**c–e**); impaired fission in (**h–j**). Time scale included per event. Blue and orange arrows point to two lysosome undergoing fusion, or one lysosome budding from a parent lysosome. In all images, scale bars represent 1 µm. See also Supplementary Fig. 4a–h and Supplementary Movie 8. **k–o** Parameters indicated in each graph were quantified from live neurons transfected and labeled as in (**a–j**) and imaged for 300 s every 1 s. pSuper control ($n = 13$), white bars and RTN4 plus DP1 knockdown ($n = 14$), gray bars. Boxplots show 25/75-percentiles, the median, and individual datapoints each represent a neuron. Whiskers represent min to max; ns—not significant, ***$p < 0.001$ and **$p < 0.01$ comparing conditions to control (two-sided Mann–Whitney $U$ (**k–o**). Source data and exact $p$ values are provided as a Source Data file.

also been shown to promote organelle fission by generating the forces required for organelle budding[35]. Since our findings support a model in which ER tubules contact lysosomes at the pre-axonal region to regulate lysosome size for proper lysosome translocation into the axon, we searched for an ER protein that could mediate this process. This protein should be enriched at the pre-axonal region and should be able to bind kinesin-1. Three ER proteins containing a kinesin-1-binding domain, protrudin, KTN1, and P180, have previously been proposed to act as membrane anchor proteins that couple organelles to kinesin-1 for organelle translocation in unpolarized cells[20,36–38]. To study whether kinesin-1-binding ER proteins are involved in lysosome translocation into the axon, we knocked down protrudin, KTN1 and P180 in neurons and analyzed lysosome distribution.

We observed that only P180 knockdown reduced lysosome distribution in the axon (Supplementary Fig. 5a–c), while the distribution of other organelles such as mitochondria, Rab11-positive recycling endosomes and Rab3-positive vesicles were not affected (Supplementary Fig. 2c, e–g, i, j). P180 knockdown reduced lysosome translocation into the axon, as the total number of moving lysosomes was decreased, while stationary lysosomes remained unaffected in the proximal axon; a phenotype similar to what we observed after ER tubule disruption (Figs. 6a, b, 1e, f). Knockdown of P180 also resulted in enlarged lysosomes that accumulated in a pre-axonal region (Fig. 6c; Supplementary Movie 9).

We previously showed that P180 is enriched in ER tubules preceding the axon initial segment[13]. P180 and kinesin-1 KIF5A-Rigor, a motor mutant that can bind to but not walk nor dissociate from MTs, co-distributed in a pre-axonal region, indicating this is their main site of interaction (Fig. 6d, e). Moreover, we observed lysosomes in contact with P180-enriched

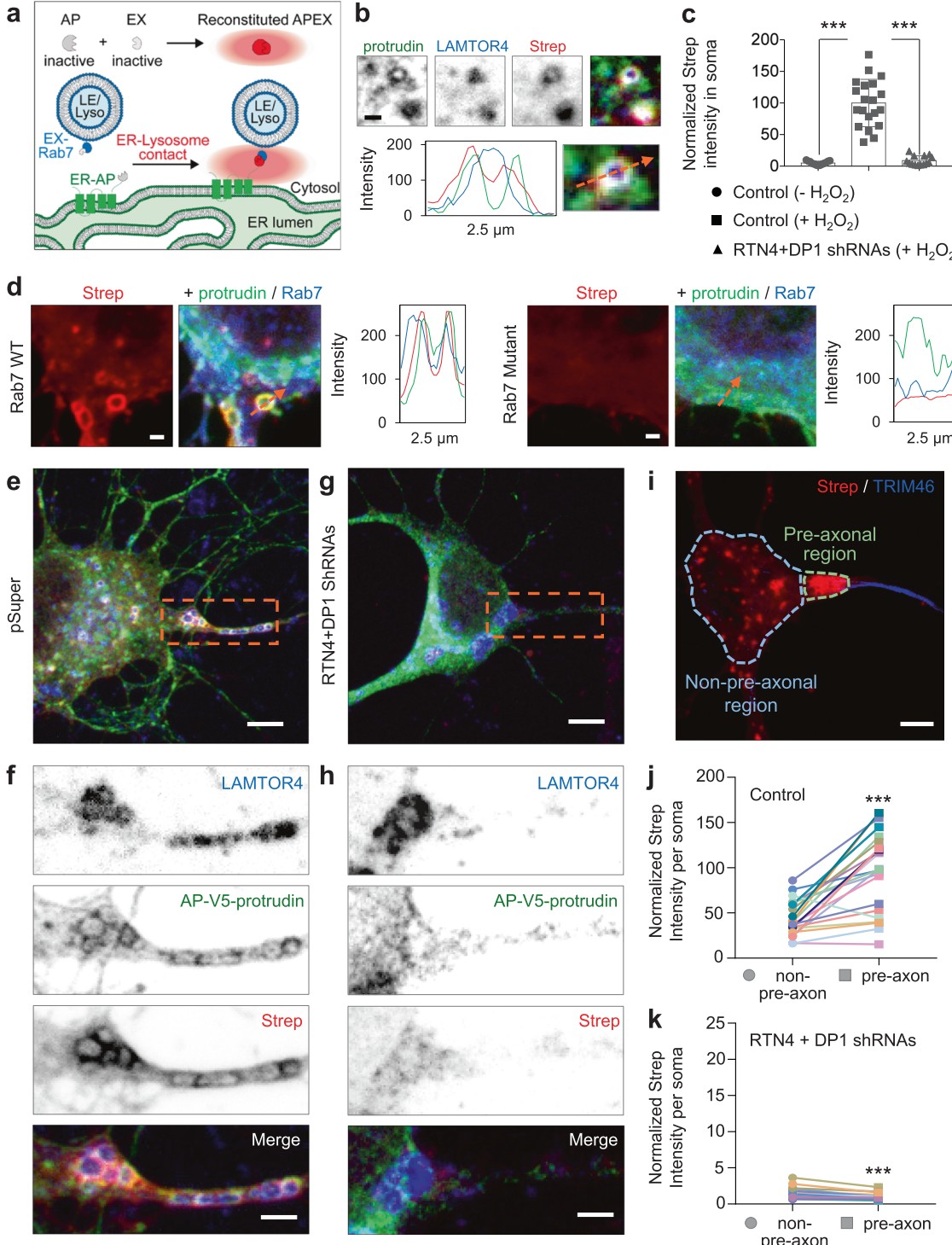

ER tubules in close proximity to kinesin-1-decorated MT tracks in this region (Fig. 6f). The enrichment of P180 in a pre-axonal region required ER tubule formation, as ER tubule disruption caused the re-distribution of P180 into somatic ER cisternae absent from a pre-axonal region (Supplementary Fig. 5d–f).

Besides a kinesin-1-binding domain (BD), P180 contains a MT-BD located in a basic decapeptide repeat domain involved in ER tubule–MT co-stabilization[13,39]. To determine whether the kinesin-1-BD and/or the MT-BD of P180 are required for axonal lysosome translocation, we performed knockdown and rescue experiments with shRNA-resistant P180 constructs (Fig. 6g). Co-expression with full-length P180 rescued the reduced

translocation of lysosomes into the axon after P180 knockdown, while co-expression with P180ΔKIF5-BD or P180ΔMT-BD deletion constructs was not sufficient to rescue this phenotype. This indicates that both domains are required for lysosome translocation into the axon (Fig. 6h, i).

Then, we studied the dynamics of lysosomes in the soma. We found that P180 knockdown reduced lysosome motility and often resulted in an agglomeration of mature lysosomes at a pre-axonal region (Supplementary Fig. 6a; Supplementary Movie 10). We also observed a significant reduction in the number and an increase in the size of mature lysosomes, as well as a reduced number of fusion and fission events and an increased fusion/

**Fig. 5 ER – lysosome contacts are enriched in a somatic pre-axonal region and they require ER tubule formation. a** Schematic representation of Split-APEX system used to visualize ER–lysosome contacts. An ER tubule-contact marker (protrudin) is fused to an AP module and the lysosome adaptor Rab7 is fused to an EX module. Only proximity of the two proteins allows reconstitution of full APEX2. After addition of biotin-phenol (BP) and $H_2O_2$, APEX2 biotinylates proteins in close proximity and this can be detected by fluorescently conjugated-streptavidin (Strep; red radius). **b** Representative images of a magnified region from the soma of a neuron transfected with AP-V5-protrudin (green) and EX-HA-Rab7, treated with biotin-phenol (BP) and $H_2O_2$, and labeled with an antibody against LAMTOR4 (blue) and Alexa568-conjugated Strep (red). Intensity profile line, bottom. **c** Average Strep intensity in soma of neurons transfected as in (**b**) plus control pSuper or shRNAs targeting RTN4 plus DP1. Control neurons were treated with BP in the absence ($n = 13$) or presence ($n = 20$) of $H_2O_2$ and knockdown neurons treated with BP plus $H_2O_2$ ($n = 14$) prior labeling with Alexa568-conjugated Strep. **d** Representative images of a magnified region from the soma of a neuron transfected with AP-V5-protrudin (green) and EX-HA-Rab7 (blue) (left panels) or with EX-HA-Rab7T22N (blue) (right panels), treated with BP and $H_2O_2$, and labeled with Alexa555-conjugated Strep (red). Intensity profile line to the right of merged images. **e**–**h** Representative images of neurons transfected, treated, and labeled as in (**b**), plus control pSuper plasmid (**e**) and (**f**) or shRNAs targeting RTN4 plus DP1 (**g**) and (**h**). Higher magnification of dashed orange boxes in (**e**) and (**g**) are shown in (**f**) and (**h**). **i**–**k** Representative image of a neuron transfected and treated as in (**e**) and labeled for Alexa555-conjugated Strep (red) plus an antibody against TRIM46 (blue). Light blue dashed lines represent the non-pre-axonal region, light green dashed lines represent the pre-axonal region. Average Strep intensity in non-pre-axonal and pre-axonal regions of neurons transfected as in (**e**) and (**g**); (**j**; $n = 23$) and (**k**; $n = 19$); respectively. Scale bars represent 1 μm (**b**) and (**d**); 2 μm in (**f**) and (**h**); 5 μm in (**e**), (**g**), and (**i**). Data are presented as mean values ± SD in (**c**) and individual datapoints each represent a neuron in (**c**); ***$p < 0.001$ comparing conditions to control (one-way ANOVA test followed by a Sidak's multiple comparisons test) in (**c**). Line graphs shows individual datapoints each represent a neuron (depicted with different colors) in (**j**) and (**k**); ***$p < 0.001$ comparing non-pre-axonal region versus pre-axonal region (two-sided Paired $t$ test in (**j**) and two-sided Wilcoxon test in (**k**). Source data and exact $p$ values are provided as a Source Data file.

fission ratio, indicating that lysosome fission is more affected (Fig. 6j; Supplementary Fig. 6b–i). Although lysosome size was increased under P180 knockdown compared to control, the increase was modest compared to ER tubule disruption. P180 knockdown caused more agglomeration of mature lysosomes in the pre-axonal region than ER tubule disruption. This is likely a consequence of the drastic reduction in lysosome motility that impairs the ability of lysosomes to complete a final step of fission and undergo new fusion events with other lysosomes. To determine whether P180 contributes to ER–lysosome contact formation, we used the split-APEX assay (Fig. 5a). However, knockdown of P180 did not reduce the Strep signal generated by AP-protrudin and EX-Rab7, while knockdown of the main tethering proteins VAPA and VAPB caused a modest reduction in Strep signal (Fig. 6k, Supplementary Fig. 6j, k). Since knockdown of organelle tethering proteins does not always result in contact disruption[17,40–43], we overexpressed P180 and VAPB. Overexpression of VAPB but not P180 induced a significant increase in Strep signal (Supplementary Fig. 6l, m), suggesting that P180 is not involved in ER tubule–lysosome contact formation. These results indicate an important local role of P180 and its MT- and kinesin-1 binding domains, at the pre-axonal region, for proper lysosome translocation into the axon.

**Visualizing dynamics of ER–lysosome contacts, lysosome fission and translocation at the pre-axonal region.** In order to get more insights into the contacts between ER tubules and lysosomes enriched in the pre-axonal region and their relationship with fission and translocation events, we studied the co-dynamics of the ER and lysosomes by labeling lysosomes with SirLyso, the ER with Sec61β and the proximal axon with TRIM46. We observed several different types of events associated to ER–lysosome interactions such as lysosome translocation along ER tubules, lysosomes in contact with the ER undergoing fission, or lysosomes undergoing fission followed by immediate translocation into the axon (Fig. 7a; Supplementary Movie 11). We also observed ER hitchhiking on motile lysosomes after lysosome fission at contact site (Fig. 7a, Supplementary Movie 11[9,44]). To overcome the diffraction limits and crowded environment of the pre-axonal region to visualize contact sites dynamics with high precision, we used a recently developed reversible contact assay[45,46] (Fig. 7b). We coupled the dimerization-dependent fluorescent modules GB and RA to Rab7 and protrudin, respectively, and we observed that the close proximity of the ER with

lysosomes reconstituted fluorescent protein labeling at contact sites (Fig. 7b, c). ER–lysosome contacts were enriched in the pre-axonal region, similar to what we observed using the split-APEX assay. We did not observe reversible interactions between the ER and lysosomes during our 120-s live cell imaging (Fig. 7c; Supplementary Movie 11), which is consistent with evidence showing a tight interaction between them[8,9]. Lysosomes that were tightly associated to the ER were observed to undergo fission and translocation at the pre-axonal region (Fig. 7d; Supplementary Movie 11). ER tubule disruption caused impaired ER–lysosome contacts together with reduced lysosome fission, while P180 knockdown reduced lysosome fission but without any apparent effect on contact site formation (Fig. 7c, d; Supplementary Movie 11). Contacts were often observed to be stable during the whole imaging period under P180 knockdown, similar to control neurons (Supplementary Movie 11). However, these stable contacts were often surrounded by agglomerated immobile lysosomes unable to finalize fission and translocate (Fig. 7d; Supplementary Movie 11).

Together, these results indicate that fission and translocation of lysosomes occur in association with ER tubules present at the pre-axonal region. ER tubule formation is required for ER–lysosome contact formation, while P180 may contribute to a final step in contact–MT stabilization for subsequent kinesin-1-powered lysosome fission and translocation, as both its MT-binding and kinesin-1-binding domains are required for axonal lysosome translocation (Fig. 7e).

## Discussion

Here we propose a model in which ER tubule – lysosome contacts at a pre-axonal region promote kinesin-1-powered lysosome fission and subsequent translocation into the axon (Fig. 7e). We show that ER shape regulates local lysosome availability in neurons. Somatic ER tubules control lysosome size and axonal translocation by promoting lysosome homo-fission. ER tubule – lysosome contacts are enriched in a pre-axonal region, where the kinesin-1-binding ER-protein P180 interacts with axonal MTs to promote kinesin-1-dependent lysosome translocation into the axon.

**Somatic ER tubule – lysosome contacts in axonal lysosome availability.** Both the ER and lysosomes play essential roles in neuronal development and maintenance, and their distribution

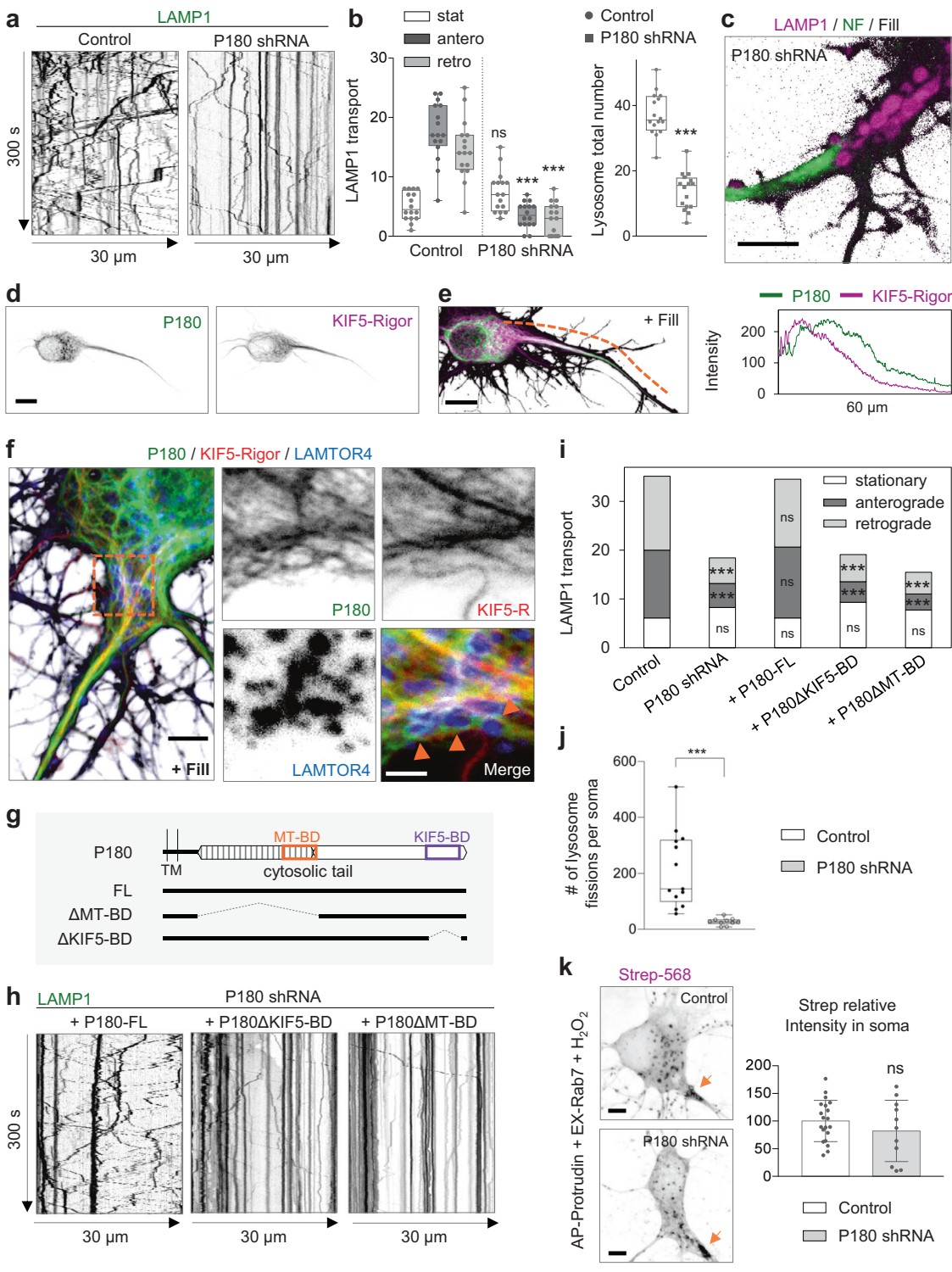

and organization must be tightly regulated to meet local demands. We have acquired a better understanding of the importance of MT-driven motor–organelle coupling for neuronal local availability of these two organelles, but only by studying each organelle in isolation[13,21]. The ER and lysosomes are both distributed along the somatodendritic and axonal domains, and contacts between these two organelles have been visualized in unpolarized cells and neurons[8,12]. How neuronal organelle availability is regulated by local organelle organization and communication via organelle–organelle contacts is a pending question. Here, we found that a balance between ER tubules and

ER cisternae is required for proper axonal distribution of lysosomes. The conversion between tubules and cisternae is regulated by ER tubule-shaping proteins such as RTNs and DP1, and the ER cisternae-shaping protein CLIMP63[14,15]. We observed that knockdown of the ER-shaping proteins RTN4 and DP1 causes a reduced lysosome distribution in the axon, while knockdown of CLIMP63 increased axonal lysosome distribution. A similar phenotype has been observed for ER distribution in neurons, in which ER tubule disruption decreases axonal ER distribution, while ER cisternae disruption increase axonal distribution of ER tubules in the axon[13]. This initially led us to speculate that axonal

**Fig. 6 The KIF5-binding and ER protein P180 is enriched in a pre-axonal region and required for axonal lysosome translocation but not for ER–lysosome contact formation. a–c** Representative kymographs of DIV7 neurons transfected with LAMP1-GFP and fill together with control pSuper plasmid or shRNAs targeting P180 in (**a**). Lysosome movement at the proximal axon was imaged for 300 s every 1 s. Quantification of lysosome movement in the proximal axon, in (**b**; n = 16 neurons per condition). Representative still image from Supplementary Movie 9 of the pre-axonal–AIS region of a neuron transfected with LAMP1-GFP (magenta), fill (gray) and stained for NF (green) in (**c**). See also Supplementary Fig. 5a–c. **d, e** Representative images of a neuron transfected with mCherry-P180 (colored green), GFP-KIF5A-Rigor (colored magenta) and fill in (**d**). Merged image and intensity profile line from dashed orange segment, in (**e**). See also Supplementary Fig. 5d–f. Image is representative of three independent experiments. **f** Representative image of a neuron transfected as in (**d, e**) and stained for LAMTOR4 (blue). mCherry-P180 (colored green), GFP-KIF5A-Rigor (colored red). Right panels show higher magnification of dashed line square. Image is representative of two independent experiments. **g** Schematic representation of P180 protein with its short luminal domain, transmembrane domain (TM), microtubule-binding domain (MT-BD in orange box) and KIF5-motor binding domain (KIF5-BD in purple box). Three constructs of P180 protein were generated as full length (FL), MT-BD-deleted construct (ΔMT-BD) and KIF5-BD-deleted construct (ΔKIF5-BD). **h, i** Representative kymographs of lysosome movement from neurons transfected as in (**a**, n = 12 per condition) together with shRNA-resistant P180-FL (n = 11), P180ΔKIF5-BD (n = 13) or P180ΔMT-BD (n = 11) constructs in (**h**). Quantification of lysosome movement in the proximal axon in (**i**). **j** Quantification of lysosome fission events from live neurons transfected and labeled as in Fig. 4 and imaged for 300 s every 1 s. pSuper control, white bars (same as in Fig. 4; n = 13) and P180 knockdown (n = 10), gray bars. Knockdown experiments performed on the same day as control neurons. See also Supplementary Fig. 6a–i and Supplementary Movie 10. **k** Representative images of neurons expressing split-APEX system and treated as in Fig. 5b, co-expressing a control pSuper vector or shRNAs targeting P180 and labeled with Alexa568-conjugated Strep. Orange arrows point to a pre-axonal region. Graph shows the relative streptavidin intensity in control neurons versus shP180 treated neurons. n = 20 and 11 neurons, respectively. See also Supplementary Fig. 6j–m. Scale bars represent 10 μm in (**d**) and (**e**), 5 μm in (**c**), (**f**, left panel) and (**k**), and 2 μm in (**f**, right panels). Individual datapoints each represent a neuron in (**b**, **j**, and (**k**). Boxplot shows 25/75-percentiles, the median and whiskers represent min to max in (**b** and **j**). Data are presented as mean values ± SD in (**k**); ns—not significant and ***p < 0.001 (one-way ANOVA followed by a Sidak's multiple comparison test) in (**b**), (Kruskal–Wallis test followed by Dunn's multiple comparison test) in (**i**), (two-tailed Mann-Whitney U) in (**j**) and (two-tailed unpaired t test) in (**k**). Source data and exact p values are provided as a Source Data file.

ER tubules contribute to the abundance of axonal lysosomes. However, axonal ER tubule repositioning into the soma did not affect the distribution or transport of lysosomes along the axon. On the contrary, we found that somatic ER tubule redistribution into the axon, caused impaired axonal lysosomes translocation. Interestingly, a previous EM study in brain tissue showed that ER – lysosome contacts mainly occur in the soma[12], a finding that we further confirmed using the split-APEX and GB-RA contact assays to visualize contact sites. Together this suggests that the importance of somatic ER tubule organization in regulating axonal lysosome translocation is mediated by ER – lysosome contacts. Indeed, we found that ER tubule organization is required to form these contacts with lysosomes. Several ER–organelle tethering proteins have been identified at contact sites, with the ER protein VAP playing a broader role in ER tethering to multiple organelles as well as the plasma membrane[7]. We analyzed the role of VAP in axonal lysosome distribution and contact formation by knockdown experiments. We found only a modest reduction in both axonal distribution and in contact formation after knockdown of both VAPA and VAPB (Supplementary Fig. 5a, b; Supplementary Fig. 6j, k), but we cannot discard a possible pleiotropic effect of VAP knockdown in our system. Knockdown of protrudin, another tethering protein[20], did not disrupt axonal lysosome distribution. Although several tethering protein pairs have been identified in ER – endosome or lysosome contact sites, their knockdown does not always result in contact loss, which suggests that other molecules may compensate for contact formation[17,40–43]. We have referred to LE and lysosomes as lysosomes, because of the use of markers present in both highly dynamic populations; however, ER – LE and ER – lysosome contact sites may or may not use different tethering proteins[20,47,48]. It therefore remains unclear from our study which specific contact site (ER – LE and/or ER – lysosome) contribute(s) to axonal LE/lysosome availability.

**ER tubules regulate lysosome size and motility.** ER – endosome contacts increase as endosomes mature into a lysosome[8]. These contacts have been shown to promote endosome fission in non-neuronal cells[19]. Here, we have shown that ER tubule disruption causes enlarged and less motile mature lysosomes. Hundreds of fusion and fission events between mature and immature lysosomes were observed in the soma of control neurons in a period of 300 s, while ER tubule disruption caused a drastic reduction of around 57% in lysosome fission events per lysosome. This indicates an important role for ER tubules in lysosome fission in order to maintain proper lysosome size and number to meet local demands in neurons.

A recent study has reported that knockdown of spastin, a MT-severing protein associated to ER tubules, also results in impaired endosome fission and enlarged lysosomes. Spastin and actin nucleators, such as the WASH complex component strumpellin, could generate the environment to promote lysosome constriction and fission at ER tubule – lysosome contact sites[24]. In the same study, they also observed increased secretion of lysosomal enzymes into the extracellular space of non-neuronal cells, suggesting impaired trafficking of enzymes into lysosomes[24]. However, we have detected enzyme activity (active Cathepsin B and D) within enlarged lysosomes in live neurons and the presence of intraluminal vesicles in enlarged lysosomes by CLEM, indicating that these membranes are subject to lysosomal degradation.

In our study, enlarged mature lysosomes were often less motile after ER tubule disruption, suggesting there may also be impaired coupling to the kinesin-1 motor. Besides its function in lysosome translocation, kinesin-1 was also shown to be involved in lysosome fission[35]. Consistent with this, we found that disruption of the kinesin-1-binding ER protein P180 caused a drastic reduction in lysosome motility and the enlargement of mature lysosomes, although they were smaller compared to ER tubule disruption. It is possible that ER–lysosome interactions stabilize the parent lysosome to facilitate transfer of the kinesin-1 motor to the budding lysosome, which can then generate the forces to complete the fission and promote its subsequent translocation.

**ER–lysosome–MT interplay at a pre-axonal region in axonal organelle translocation.** Interestingly, we observed a striking enrichment of contacts between the ER and lysosomes at a pre-axonal region. This region is featured by the landing of the kinesin-1 motor on stable MTs, where it is required for lysosome and ER tubule translocation into the axon[13,21,34]. We previously

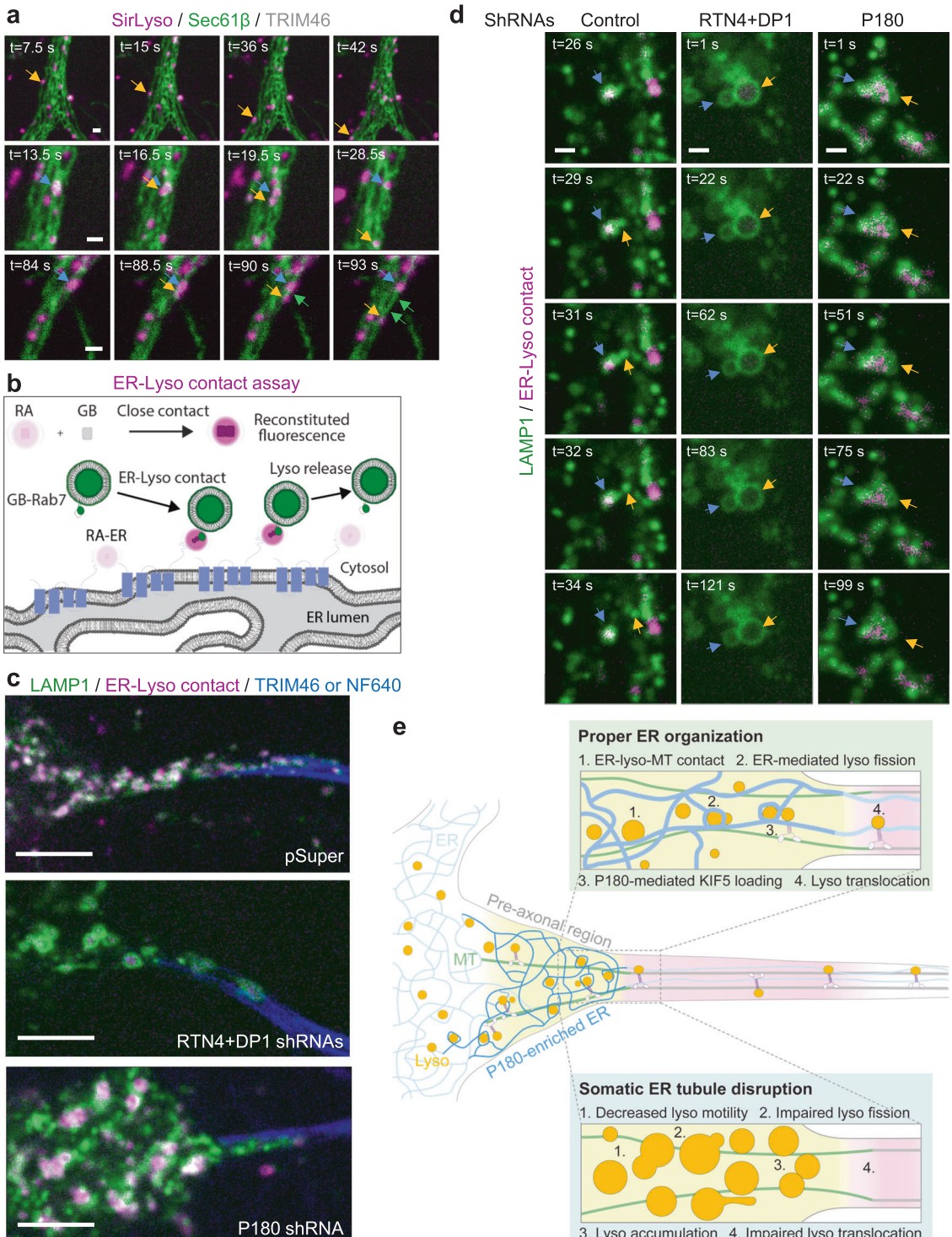

**Fig. 7 Lysosome fission and translocation associated to ER – lysosome contact sites at the pre-axonal region. a** Representative still images of DIV6 neurons co-transfected with Sec61β-GFP (green) and TRIM46-BFP and labeled for lysosomes with SirLyso (magenta) prior to imaging for 120 s every 1.5 s. Lysosome translocation along an ER tubule (top panels); Lysosome fission at contact sites followed by lysosome translocation (middle panels); lysosome fission and co-translocation with an ER tubule (bottom panels). See also Supplementary Movie 11. **b** Schematic representation of the reversible GB/RA contact assay used to visualize ER – lysosome contacts in live cells. GB and RA modules were tagged to RAB7 and protrudin, respectively. Only proximity of the two proteins enables reconstitution of fluorescent protein at contact sites (magenta). **c** Representative still images of ER–lysosome contacts (magenta) at pre-axonal region from neurons co-transfected with control pSuper or shRNAs targeting RTN4 plus DP1 or shRNAs targeting P180 together with LAMP-GFP, GB-Rab7, RA-Protrudin, and TRIM46-BFP (blue) or labeled with NF640 (blue) prior imaging. **d** Representative still images of fusion events from neurons co-transfected and labeled as in (**c**). **e** Schematic representation of our proposed model of ER tubules regulating axonal lysosome availability at the pre-axonal region. Scale bars represent 1 μm in (**a**) and (**d**) and 5 μm in (**c**).

found that the ER protein P180 is enriched in axonal ER tubules at a region preceding the axon initial segment, and it is involved in ER–MT co-stabilization[13]. Here we show that, in this same region, P180 associates with stable MTs decorated by the kinesin-1 KIF5A-rigor mutant and that P180 is required for lysosome translocation into the axon. We observed an ER ring rearrangement around lysosomes at a pre-axonal region. ER rings around lysosomes have previously often been observed interacting with MTs, and they reduce diffusive motility of lysosomes[8,9]. In the absence of contacts with the ER, even when bound to MTs, lysosomes tend to undergo diffusive movement rather than directional transport along MTs[9]. Guo et al. proposed that ER–lysosome interactions may assist in stabilizing lysosomes prior their docking onto MTs via molecular motors[9]. Consistent with a possible role of P180 in this process, we observed a drastic impairment in lysosome directional motility in P180 knockdown neurons. The cytoplasmic tail of P180 contains a MT-BD in a basic decapeptide repeat region, and a KIF5-BD in a coiled-coil (CC) region at the end of the C-terminal tail[38,39]. In neurons, expression of a P180ΔCC deletion construct containing the MT-BD, but not the KIF5-BD, promotes ER tubule–MT co-stabilization and distribution of P180 along the axon[13]. In this study, we find that both the MT-BD and KIF5-BD are required for proper lysosome translocation into the axon. The role of P180 is likely downstream of ER tubule formation and ER tubule–lysosome contacts formation, as knockdown of P180 did not result in an evident reduction in contact formation. Concordantly with a previous finding of kinesin-1-mediating lysosome fission[35], we observed that P180 disruption produces an agglomeration of less motile lysosomes mainly in a pre-axonal region, which were unable to separate from each other and translocate into the axon. We propose a multi-step model, in which the MT-BD of P180 locally stabilizes the interaction of ER tubule–lysosome contacts with MTs at a pre-axonal region. Then, the kinesin-1-BD of P180 facilitates kinesin-1 loading onto the budding lysosome, while part of the lysosome remains stabilized by the ER–MT interaction. Loading of kinesin-1 onto the budding lysosome at a pre-axonal region could locally facilitate the final step in lysosome fission and promote its subsequent translocation into the axon (Fig. 7e). In addition to this, P180 is known to play a role in both ribosome binding and mRNA localization to the ER[49], but it is unlikely that this function is directly important for ER–lysosome–MT interactions.

Other studies in cell lines have shown that the ER-tethering protein protrudin, which also contains a kinesin-1-binding domain, promotes kinesin-1 loading onto lysosomes[20]. We find that protrudin is distributed in ER tubules wrapping lysosomes at a pre-axonal region; however, knockdown of protrudin did not impair lysosome translocation into the axon, suggesting that other ER proteins may compensate in promoting lysosome translocation into the axon. In addition, it remains unknown whether P180 or other proteins enriched in contact sites at the pre-axonal region bind to any adaptor protein on the lysosomal membrane for proper ER-mediated lysosome translocation into the axon. The lysosomal adaptor complex BORC–Arl8–SKIP–KLC is required for lysosome translocation into the axon and SKIP–KLC has been shown to strongly bind MTs decorated by kinesin-1 rigor at the pre-axonal region[21], which co-distributes with P180. Recently, Arl8b and SKIP were shown to be required for ER tubule hitchhiking on motile lysosomes[44]. We have also observed ER hitchhiking on translocating lysosomes at the pre-axonal region, which suggest that the same tethering proteins may be involved in lysosome and ER translocation into the axon.

Together, our results support a model in which ER tubule–lysosome contacts interact with stable axonal MTs at a pre-axonal region to locally promote kinesin-1-powered lysosome

fission and subsequent kinesin-1-mediated translocation into the axon. More broadly, our results suggest that organelle organization, inter-organelle communication and organelle transport are finely orchestrated to control local organelle availability in neurons. The fact that several ER-shaping proteins and contact tethering proteins are mutated in the neurodegenerative diseases hereditary spastic paraplegia and amyotrophic lateral sclerosis, highlight the importance of ER organization and inter-organelle communication in neuronal health[17,43].

## Methods

**Animals**. All experiments were approved by the DEC Dutch Animal Experiments Committee (Dier Experimenten Commissie), performed in line with institutional guidelines of University Utrecht, and conducted in agreement with Dutch law (Wet op de Dierproeven, 1996) and European regulations (Directive 2010/63/EU). The animal protocol has been evaluated and approved by the national CCD authority (license AVD1080020173404). Female pregnant Wistar rats were obtained from Janvier, and embryos (both genders) at embryonic (E)18 stage of development were used for primary cultures of hippocampal neurons. The animals, pregnant females and embryos have not been involved in previous procedures.

**Primary neuronal cultures and transfection**. The hippocampi from embryonic day 18 rat brains were dissected and dissociated in trypsin for 15 min and plated on coverslips coated with poly-L-lysine (37.5 μg/ml) and laminin (1.25 μg/ml) at a density of 100,000/well or 50,000/well (12-well plates) to prepare primary hippocampal neurons. The day of neuron plating corresponds to day-in-vitro 0 (DIV0). Neurobasal medium (NB) supplemented with 1% B27 (GIBCO), 0.5 mM glutamine (GIBCO), 15.6 μM glutamate (Sigma), and 1% penicillin/streptomycin (GIBCO) was used to maintain the neurons incubated under controlled temperature and $CO_2$ conditions (37 °C, 5% $CO_2$). Hippocampal neurons were transfected using Lipofectamine 2000 (Invitrogen). Briefly, DNA (0.05–2 μg/well) was mixed with 1.2 μl of Lipofectamine 2000 in 200 μl Opti-MEM, incubated for 20 min at room temperature, then added to neurons in NB and incubated for 1 h at 37 °C in 5% $CO_2$. Next, neurons were washed with NB and transferred to their original medium at 37 °C in 5% $CO_2$ until fixation or imaging at different days in vitro (DIV) as indicated.

**DNA and shRNA constructs**. The following vectors were used: pEGFP(A206K)-N1 and pEGFP(A206K)-C1 (a gift from Dr. Jennifer Lippincott-Schwartz), pGW1-mCherry and pGW1-BFP[50] and pSuper[51]. GFP-KIF5A-Rigor, LAMP1-GFP and mCherry-KIF5A-motor-Strep were a gift from Dr. Juan Bonifacino[21,34] and RFP-CLIMP63 was a gift from Dr. Tom Rapoport. RTN4A-GFP was provided by Dr. Gia Voeltz[52] (Addgene plasmid #61807). V5-GFP-P180 full length[53] (Addgene #92150), TOM20-V5-FKBP-split-AP and Split-EX-HA-FRB-CB5[32] (Addgene #120914 and #120915, respectively) were provided by Dr. Alice Ting. GB-NES and RA-NES[45] (Addgene #61017 and #61019, respectively) were provided by Dr. Robert Campbell. LAMP1-RFP was provided by Dr. Walther Mothes[54] (Addgene #1817). GFP-Rab7a and GFP-Rab11a were previously described[55]. Mito-DsRed[56], TRIM46-BFP[13], VAPB-GFP and GFP-Rab3A (Dr. Casper Hoogenraad, unpublished data), were used. For Strep/SBP heterodimerization system, the cloning of Strep-KIFC1-MD-HA and GFP- or mCh-SBP-RTN4A has been previously described[13]. P180ΔMT-BD-GFP construct corresponds to a deletion construct lacking the entire P180 decapeptide repeat domain (containing the MT-BD), and it was previously described (named as P180-Δrepeat-GFP in[13]. We were unable to generate a deletion construct lacking only the MT-BD because of the nature of the repeated decapeptide sequence present in this domain of P180.

All the primers used in this study are provided in Supplementary Table.
The plasmids generated in this study include:
For HA-KIF5A-Strep, the mCherry sequence from mCherry-KIF5A-motor-Strep[34] was removed by digestion with AgeI and BsrGI enzymes and replaced by a 3x HA sequence.
For P180-mCherry, full length P180 was PCR amplified from V5-GFP-P180 (Addgene #92150) and inserted in mCherry-N1 vector between XhoI and BamHI sites. A 3x(glycine-serine) linker was generated by addition to the cloning primers and was introduced between P180 and before the mCherry sequence to allow freedom of movement between domains.
For the P180 deletion construct P180-ΔKIF5-BD-GFP, DNA sequences between nucleotides 1-3877 and 4236-4617 were PCR amplified from V5-GFP-P180 (Addgene #92150) and the two fragments were assembled and cloned into pEGFP(A206K)-N1 between XhoI and BamHI sites by GIBSON assembly. A 3x (glycine-serine) linker was introduced between fragments and before the GFP sequence.
For the Split APEX assay, we generated Split-AP-V5-protrudin and Split-EX-HA3x-Rab7a as follows: First, the GFP sequence in GFP-C1 vector was removed and replaced with Split-AP-V5 and Split-EX-HA3x sequence between AgeI and BglII sites. To generate Split-AP-V5-C1 vector, V5 and AP fragments were amplified from GFP-V5-P180[53] (Addgene # 92150) and TOM20-V5-FKBP-split-

AP[32] (Addgene# 120914), respectively. A 3x(glycine-serine) linker was introduced before and after the V5 sequence. To generate the Split-EX-HA3x-C1 vector, EX and HA fragments were amplified from Split-EX-HA-FRB-CB5[32] (Addgene# 120915) and HA3x-KIF5A-Strep (generated in this study), respectively. A 3x (glycine-serine) linker was introduced between EX and HA fragments. Then, to generate Split-AP-V5-protrudin and Split-EX-HA-Rab7, human protrudin sequence was PCR amplified from IMAGE 4818199 (SourceBioScience) and Rab7a sequence was PCR amplified from GFP-Rab7[55]. Both protrudin and Rab7 were cloned into Split-AP-V5-C1 and Split-Ex-ln-HA3x-C1 vectors, respectively, between XhoI and EcoR1 sites by GIBSON assembly. A 3x(glycine-serine) linker was introduced before the Rab7a sequence. To generate Split-EX-HA3x-Rab7a-T22N mutant construct, a T22N mutation was inserted by replacing ACA with AAT by using the QuickChange II XL Site-Directed Mutagenesis Kit.

For the GB/RA system, we generated GB-V5-Rab7 and RA-HA-protrudin as follows: First, the GFP sequence in the GFP-C1 vector was removed and replaced with GB-V5 and RA-HA sequences between AgeI and BsrGI sites. To generate GB-V5-C1 vector, GB and V5 fragments were amplified from GB-NES[45] (Addgene #61017) and Split-AP-V5-C1 vector, respectively. To generate RA-HA-C1 vector, RA and HA fragments were amplified from RA-NES[45] (Addgene #61019) and Split-EX-HA-C1 vector, respectively. Then to generate GB-V5-Rab7 and RA-HA-protrudin, canine Rab7a and human protrudin sequences were amplified as explained above. Rab7 was cloned into GB-V5-C1 vector between BglII and SalI sites and protrudin was cloned into RA-HA-C1 between XhoI and EcoR1 sites, by GIBSON assembly.

The following sequences for rat-shRNAs, inserted in pSuper vector, were used in this study: RTN4-shRNA (5′-gtccagatttctctaatta-3′) and DP1-shRNA (5′-gacatataaagt tccagaa-3′) validated in ref. [13]; P180-shRNAs (5′-tcagtgcaattgtctgtat-3′ and 5′-taaaccaa ccaacacagcg-3′) used in ref. [13] and validated in Supplementary Fig. 5C; KTN1-shRNA (5′-ggaccttctcaagaggtta-3′) and CLIMP63-shRNA (5′-tcaaccgtattagtgaagttctaca-3′) validated in ref. [57], and sequence previously used in ref. [13]; VAPA-shRNA (5′-gcat gcagagtgctgtttc-3′) and VAPB-shRNA (5′-ggtgatggaagagtgc-3′) validated in ref. [58], 2007 and used in ref. [22]; protrudin-shRNA (5′-aagcttcttgatccgactggaag-3′) validated in ref. [59], and sequence cloned into pSuper vector after oligo annealing.

**Antibodies and reagents**. The following primary antibodies were used in this study: rabbit anti-LAMTOR4 (Cell Signaling, clone D6A4V, Cat# 12284S, RRID: AB_2797870, 1/250), mouse anti-EEA1 (BD Biosciences, Cat# 610456, RRID: AB_397829, 1/200), mouse anti-P62 (Abcam, Cat# 56416, RRID:AB_945626, 1/500), mouse anti-V5 (Thermo Fisher Scientific Cat# R960-25, RRID:AB_2556564, 1/1000), rat anti-HA (Roche Cat# 11867423001, RRID:AB_390918, 1/200), mouse anti-Pan-Neurofascin external (clone A12/18; UC Davis/NIH NeuroMab, Cat# 75-172, RRID: AB_2282826, 0.18 mg/ml), rabbit anti-TFE-3 (Cell Signaling, Cat#14779, RRID:AB_2687582, 1/250), mouse anti p62/SQSTM1 (Abnova, Cat# H00008878-M01, RRID:AB_437085, 1/500) and rabbit anti-TRIM46[60] (1/500), mouse-anti-GFP (Thermo Fisher Scientific Cat# A-11120; RRID:AB_221568, 1/250), rabbit-anti-RRBP1/P180 (Abcam; Cat#ab95983, RRID:AB_10678752, 1/500), mouse-anti-alpha-tubulin (Sigma, clone B-5-1-2, Cat#T5168, RRID:AB_477579, 1/10,000).

The following secondary antibodies were used in this study: Strep, Alexa Fluor-555 conjugate (Thermo Fisher Scientific Cat# s21381, RRID: AB_2307336, 1/2000), Strep, Alexa Fluor-568 conjugate (Thermo Fisher Scientific Cat# S-11226, RRID: AB_2315774, 1/1000), donkey anti-mouse Alexa488 (Molecular Probes, Cat# A21202, RRID: AB_141607, 1/1000), goat anti-Mouse IgG (H + L) Highly Cross-Adsorbed Alexa Fluor 488 (Thermo Fisher Scientific Cat# A-11029, RRID: AB_2534088, 1/1000), donkey anti-mouse Alexa555 (Molecular Probes, Cat# A31570, RRID: AB_2536180, 1/1000), donkey anti-mouse Alexa647 (Molecular Probes, Cat#A31571, RRID: AB_162542, 1/1000), donkey anti-rabbit Alexa488 (Molecular Probes, Cat# A21206, RRID: AB_141708, 1/1000), donkey anti-rabbit Alexa555 (Molecular Probes, Cat# A31572, RRID: AB_162543, 1/1000), donkey anti-rabbit Alexa647 (Molecular Probes, Cat# A31573, RRID: AB_2536183, 1/1000), goat anti-mouse Alexa405 (Molecular Probes, Cat# A31553, RRID: AB_221604, 1/500), goat anti-rabbit Alexa405 (Molecular Probes, Cat# A31556; RRID: AB_221605, 1/500), goat anti-rat Alexa 488 (Thermo Fisher Scientific Cat# A-11006, RRID:AB_2534074, 1/1000), goat anti-rat Alexa 568 (Thermo Fisher Scientific Cat# A-11077, RRID:AB_2534121, 1/1000), IRDye 680RD goat anti-mouse IgG antibody (LI-COR Biosciences, Cat#926-68070, RRID:AB_10956588, 1/15000), IRDye 800CW goat anti-rabbit IgG antibody (LI-COR Biosciences, Cat#926-32211, RRID:AB_621843, 1/15000).

Other reagents used in this study were NeutrAvidin (Thermo Fisher Scientific, Cat# 31000), Lipofectamine 2000 (Invitrogen, Cat#1639722), SiR-lysosome kit (Spirochrome, Cat# SC012), Magic Red (ImmunoChemistry Technologies, Cat# 937); antibody labeling kit Mix-n-Stain CF640R (Biotium); heme (Sigma-Aldrich, Cat#51280); biotin-phenol (Iris Biotech, Cat#LS.3500); H2O2 (Sigma-Aldrich, Cat#H1009), DAPI (Invitrogen, Cat#: D1306), QuikChange II XL Site-Directed Mutagenesis Kit (Agilent, Cat#200521), acryloyl X-SE (AcX) (Thermo Fisher, Cat# A20770).

**Immunofluorescence staining and imaging**. Neurons were incubated at RT with pre-warmed 4% paraformaldehyde plus 4% sucrose in PBS for 20 min for fixation. Then, cells were permeabilized with 0.2% Triton X-100 in PBS supplemented with calcium and magnesium (PBS-CM) for 15 min, followed by blocking with 0.2%

porcine gelatin in PBS-CM for 30 min at 37 °C. Next, neurons were incubated with primary antibodies and then with secondary antibodies for 30 min at 37 °C each. After incubation with primary and secondaries antibodies, the cells were washed with PBS-CM three times for 5 min each. For DAPI staining, cells were incubated with 0.1 μg/ml DAPI diluted in PBS for 5 min and were washed with PBS-CM three times for 5 min each, prior to mounting. Coverslips were mounted in Fluoromount-G Mounting Medium (ThermoFisher Scientific). Only cells displaying continuous labeling of the cytosolic fluorescent protein (fill) along the somatodendritic and axonal domains, were imaged by using a confocal laser-scanning microscope (LSM700, with Zen imaging software (Zeiss) version 8.1.7.484) equipped with Plan-Apochromat ×63 NA 1.40 oil DIC and EC Plan-Neofluar ×40 NA1.30 Oil DIC objectives.

**Labeling mature lysosomes**. Prior to live-cell imaging, DIV7 hippocampal neurons were incubated with SirLyso (1000 nM in NB; Spirochrome) to detect cathepsin D activity, or Magic-Red (1:250 dilution in NB from recommended stock reconstruction; ImmunoChemistry Technologies) to detect cathepsin B activity. Both probes were incubated for 30 min under controlled temperature and CO2 conditions (37 °C, 5% CO2). After washing twice with NB, cells were supplemented with their original medium and immediately imaged.

**Correlative light and electron microscopy**. For correlation of FM and 3D-EM of neurons, FM imaging was performed prior to sample preparation for EM. Neurons were cultured on carbon-coated, gridded coverslips. DIV7 neurons incubated with SirLyso were rinsed and fixed with 4% paraformaldehyde plus 4% sucrose in 0.1 M PB for 120 min. Coverslips were imaged in fixative solution by using a confocal laser-scanning microscope (LSM700, Zeiss) equipped with Plan-Apochromat ×63 NA 1.40 oil DIC objective. The position of cells relative to the pattern etched in the coverslip was registered using polarized light. After fluorescent imaging, neurons on coverslips were post-fixed with 1% OsO4 with 1.5% K4Fe(II)(CN)6 in 0.1 M PB for 1 h on ice, followed by washing steps in ddH2O. Cells were stained with 2% uranyl acetate in ddH2O at room temperature, followed by further washing steps with ddH2O. Finally, samples were subjected to a graded ethanol series for dehydration. After dehydration, samples were flat embedded in Epon resin (ratio: 12 g Glycid Ether 100, 8 g dodecenylsuccinic anhydride, 5.5 g methylnadic anhydride, 560 μL N-benzyldimethylamine). After Epon polymerization, the resin blocks were removed from the coverslips and prepared for EM as reported before[29] with slight modifications. Regions of interest selected based on fluorescent imaging (LAMP1-GFP and SirLyso) were cut out using a clean razor blade, and glued to empty Epon sample stubs, with the basal side of the cells facing outwards. The resin-embedded neurons were then mounted on aluminum SEM stubs using carbon adhesive, and the sides of the block were covered with conductive carbon paint. Samples were imaged using a Scios Dualbeam FIB-SEM (Thermo Fischer Scientific) under high vacuum conditions. A 500 nm-thick Pt layer was deposited over the ROI using the FIB (30 kV, 1 nA). Then the trenches around the selected ROI were milled, and the imaging surface was polished. Automated serial imaging was performed using Slice&View v3 (Thermo Fischer Scientific), at low acceleration voltages (2 kV) using 5 nm pixel size, dwell time 5 μs at a slice thickness of 5 nm providing iso-tropic pixels in 3D. Backscattered electrons were collected using the in-lens backscatter detector operating in 'Optitilt' mode. Images were saved as separate 8-bit TIFF files.

The resulting images were imported in Fiji (Fiji is just ImageJ) to generate 3D volumes as a single stack and aligned using Fiji Plugin SIFT. Aligned XZ stacks were reconstructed as XY stacks (FM imaging plane) and saved as a single TIFF. Aligned and reconstructed slices were manually registered over fluorescent images. For correlation of FM and EM data, the best matching XY plane from the reconstructed stack of the ROI was overlayed with FM data using Photoshop. Multiple corresponding spots (e.g. lysosomes) on images were selected and overlay of FM and EM data was generated by linear scaling and transformation steps were followed. Only linear transformation options were used to achieve the overlays shown in the Fig. 3k.

**Live-cell imaging**. For live-cell imaging experiments, an inverted microscope Nikon Eclipse Ti-E (Nikon), equipped with a Plan Apo VC ×100 NA 1.40 oil and a Plan Apo VC ×60 NA 1.40 oil objective (Nikon), a Yokogawa CSU-X1-A1 spinning disk confocal unit (Roper Scientific), a Photometrics Evolve 512 EMCCD camera (Roper Scientific) or Photometrics Prime BSI camera, and an incubation chamber (Tokai Hit) mounted on a motorized XYZ stage (Applied Scientific Instrumentation) was used. MetaMorph (Molecular Devices) version 7.10.2.240 software was installed for controlling all devices. Coverslips mounted in a metal ring and supplemented in the original medium from neurons were imaged in an incubation chamber that maintains optimal temperature and CO2 (37 °C and 5% CO2). To visualize proteins with a specific fluorescent tag for single-color acquisition, a laser channel was exposed for 100–200 ms while for dual-color acquisition, different laser channels were exposed for 100–200 ms sequentially. Neurons were imaged every 1 or 1.5 s for 120 or 300 s. To identify the axon, neurons were co-transfected with the AIS marker TRIM46-BFP[13,60] or incubated with a CF640R-conjugated antibody against the AIS protein neurofascin (NF-640R)[61] for 30 min before live-

cell imaging. Total time and intervals of imaging acquisition for each experiment are depicted in each legend for figure and/or legend for Supplementary Movie.

**Expansion microscopy.** Immunofluorescence was performed as described above with some minor changes to increase signal retention in the expanded samples. Primary antibodies were incubated overnight at 4 °C at double the concentration. Secondary antibodies were incubated for 2–3 h at room temperature also at double the concentration. After immunofluorescence, the TREx expansion protocol was applied to expand the samples[25]. Briefly, cells were treated with 0.1 mg/ml acryloyl X-SE (AcX) in PBS overnight at room temperature. The gelation solution was prepared, containing 1.1 M sodium acrylate, 2.0 M acrylamide, 90 ppm N,N′-methylenebisacrylamide (bis), PBS (1×), 1.5 ppt APS, and 1.5 ppt TEMED. The gelation solution was placed onto a parafilm covered slide in a silicon mold, after which the sample coverslips were placed on top. The samples were allowed to polymerize for 1 h at 37 °C. Then, they were transferred to the digestion buffer (7.5 U/ml Proteinase-K in TAE buffer supplemented with 0.5% triton-X-100 and 0,8 M guanidine–HCl) and incubated for 5 h at 37 °C. Afterwards the gels were transferred to a Petri-dish and water was added to start the expansion process. Water was exchanged after an initial 30 min of expansion and following subsequent overnight expansion. After full expansion, part of the gel was excised using a scalpel and mounted on 1.5 thickness coverslips coated with PLL.

Images were acquired using a Leica TCS SP8 STED ×3 microscope equipped with a HC PL APO ×86/1.20 W motCorr STED (Leica 15506333) water objective. For excitation a pulsed white laser (80 MHz) was used. After imaging all data was deconvolved using Huygens Professional (SVI) version 20.04. and further processed using ImageJ 1.53c and Arivis 3.4.

**Strep/SBP heterodimerization system assay.** Controlled coupling between MT-driven motor proteins and a specific cargo such as vesicles, lysosomes, and ER tubules, using the Strep/SBP heterodimerization system[13,21,34]. Neurons were transfected at DIV5 with Strep-KIFC1-MD-HA plus GFP-SBP-RTN4A to pull axonal ER tubules to soma (Fig. 2b, d; Supplementary Movies 4 and 6) or HA-KIF5A-Strep plus GFP-SBP-RTN4A to pull ER tubules from the soma into axon (Fig. 2c, d, 3c; Supplementary Fig. 2b, d, h, 3b–f, i (bottom panels); Supplementary Movies 4 and 6). Strep-SBP uncoupling was prevented by adding NeutrAvidin (0.3 mg/ml) to the cell medium after 1 h of transfection[61].

**Split APEX assay.** Neurons were transfected at DIV4 with AP-V5-Protrudin and EX-3xHA-Rab7 constructs. At DIV7, a final concentration of 6uM heme (Sigma-Aldrich) was added to the medium and after 60 min neurons were washed once with NB and 500 μM biotin-phenol (Iris Biotech) in NB with supplements was added to the neurons for 30 min. Then, proximity labeling was initiated by adding $H_2O_2$ (Sigma-Aldrich) to a final concentration of 1 mM for 1 min after which the labeling reaction was stopped by removing the medium and washing once with quenching buffer (5 mM Trolox (Sigma-Aldrich) and 10 mM sodium ascorbate (Sigma-Aldrich) in HBSS) containing 10 mM sodium azide (Merck) and twice with quenching buffer without sodium azide for 3–5 min each. Neurons were subsequently fixed and stained as described above.

**Quantification of total protein levels in primary neurons.** Total protein levels in DIV4 primary cortical neurons nucleofected at DIV0 were obtained by re-analyzing the quantitative proteomics dataset from Farías et al., 2019 (ProteomeXchange dataset PXD012264)[13]. Average TMT ratios (pSuper versus untransfected cells and RTN1/2/3/4 KD or DP1 KD versus pSuper transfected cells) for the selected lysosomal (Supplementary Fig. 3k) or autophagy-related (Supplementary Fig. 4e) proteins were used to generate dot plots in R software. As indicated, the size and color of each circle reflect TMT ratios.

**Western blot analysis of P180 protein levels.** Rat INS-1 823/3 Insulinoma cells (Merck) were cultured in RPMI 1640 GlutaMAX (TM) medium (Thermo Fisher Scientific) supplemented with fetal bovine serum (GIBCO), 50 μM β-mercaptoethanol (Sigma), and 1% penicillin/streptomycin (GIBCO). The cells were maintained at 37 °C in 5% $CO_2$. INS-1 cells in a six-well plate were transfected using Lipofectamine 2000 (Invitrogen) with an empty pSuper control plasmid or pSuper plasmids containing two different shRNAs against P180 (#1 and #2). Briefly, DNA (2 μg/well) was mixed with 5 μl of Lipofectamine 2000 in 400 μl Opti-MEM, incubated for 20 min at room temperature, then added to the cells in Opti-MEM and cells were incubated for 4–6 h at 37 °C in 5% $CO_2$. Next, cells were washed twice with PBS, RPMI 1640 culture medium with additives (as above) was added and cells were incubated at 37 °C in 5% $CO_2$. After 48–72 h, cells were washed once in ice-cold PBS and collected and lysed in RIPA buffer containing protease inhibitors cocktail (Roche) for 30 min at 4 °C. Lysates were then spun at 4 °C at 13,200 rpm for 15 min and the supernatant was collected and used for Western blot analysis. Protein lysates were resolved by SDS–PAGE on a 9% Bis-Acrylamide (Bio-Rad) gel and transferred to a nitrocellulose membrane (Bio-Rad). The blots were blocked in blocking buffer (5% skimmed milk in TBS-T) and then incubated with primary antibodies in blocking buffer overnight at 4 °C. After three washes (5 min each) with TBS-T, the blots were incubated with LI-COR secondary antibodies in blocking buffer for 1 h at RT, washed again for three times in TBS-T

and once in TBS and developed on an Odyssey CLx imaging system (LICOR) with Image Studio version 5.2 software. Images from Western blot detection from three independent experiments were imported into FIJI and RRBP1/P180 protein levels were measured and normalized to alpha-tubulin protein levels.

**Image analysis and quantification.** We identified axons and dendrites based on the length of the axons (defined as at least three times longer than dendrites) by using a fill and/or an axon initial segment marker such as TRIM46 or Neurofascin that are absent in dendrites[13,21,34,60]. Images were recorded and analyzed from 3 to 5 independent experiments. No specific strategy for randomization and/or stratification was employed. Data was analyzed for at least two people in a blind fashion by using Image J 1×/Fiji[62].

*Fluorescence line intensity plots.* The co-distribution of different markers was analyzed using ImageJ. Plot profiles were generated from lines traced along lysosomes (Figs. 3d, 5b, d; Supplementary Figs. 2a, 3g, h, 4a, b, c), or segmented line traced from a somatic pre-axonal region to the proximal axon (Fig. 6e; Supplementary Fig. 5d, e, f). The length of traced line is indicated in each intensity plot.

*PI of lysosomal markers.* We used a well-established method to quantify polarized distribution of proteins/organelles in neurons, called PI[26]. Quantification of PI was performed using ImageJ[13]. Segmented lines were drawn along three dendrites and one portion of the axon of ~200 μm (excluded the axon initial segment) in each image. Mean intensities in these areas were measured by ImageJ. After averaging the mean intensities from the three dendrites, following formula was applied to calculate the PI: in which $I_d$ is the average intensity of the three dendrites and $I_a$ is the intensity of axon. PI < 0 indicates axonal distribution, PI > 0 indicates dendritic distribution and PI = 0 stands for non-polarized distribution where $I_d = I_a$ (Fig. 1b, d; Supplementary Figs. 2g–j and 5b).

*Kymograph analysis.* Kymographs from live cell images were made using Image J[61]. Segmented lines were drawn along a 30-μm segment of the axon from the most distal part of the axon initial segment as indicated in schematic in Fig. 1F. Then regions were strengthened and re-sliced followed by z-projection to obtain kymograph. Anterograde movements were oriented in all kymographs from left to right. Time of recording and length of segments are indicated in each kymograph (Figs. 1e, 2d, 6a, h). Number of events for antero- and retrograde lysosome movement as well as for stationary and total number of lysosomes were obtained from kymographs from many cells (Figs. 1f, 2e, f, 6b, i).

*Quantification of number and size of lysosomes.* The number of LAMP1-positive lysosomes and LAMP1/SirLyso-positive mature lysosomes as well as the size of mature lysosomes was analyzed using ImageJ. The number of lysosomes were counted manually from the first frame of live soma images of 13–14 different neurons per condition by three independent observers. In total, we counted 2223 LAMP1-positive and 1588 SirLyso-positive lysosomes from 14 different control neurons, 1132 LAMP1-positive and 643 SirLyso-positive lysosomes from 14 different RTN4/DP1 KD neurons and 1065 LAMP1-positive and 717 SirLyso-positive lysosomes from 13 different P180 KD neurons. We plotted the average per neuron in Fig. 3e and Supplementary Fig. 6b. We calculated the ratio of mature/immature lysosomes by dividing the total amount of SirLyso-positive lysosome per soma by the total amount of LAMP1-positive lysosomes per soma (Fig. 3f and Supplementary Fig. 6c). To measure lysosome size, straight lines were traced along the diameter of spherical lysosomes from images of the soma from live neurons. The largest lysosome per soma was measured and averaged per condition (Fig. 3i and Supplementary Fig. 6f). We considered lysosomes with a size bigger than 1 μm as enlarged lysosomes[28]. The total number of enlarged LAMP1/SirLyso-positive lysosomes were counted manually from the first frame of live soma images. We plotted the average per neuron in Fig. 3g and Supplementary Fig. 6d. The percentage of enlarged lysosomes was calculated by dividing the number of enlarged lysosomes (>1 μm) by the total number of LAMP1/SirLyso-positive lysosomes per soma (Fig. 3h and Supplementary Fig. 6e). We used the same control neurons for comparisons with RTN4/DP1 KD neurons or P180 KD neurons in all analyses as the experiments were performed together. We also quantified number of lysosomes per soma, number of enlarged lysosomes per soma, percentage of enlarged lysosomes and largest lysosome diameter for axonal ER tubule removal, somatic ER tubule redistribution into the axon and control (17–19 neurons), as explained above, for Supplementary Fig. 3c–f.

*Quantification of lysosome fusion and fission events, and lysosome motility.* Homo-fusion and homo-fission events were analyzed using ImageJ. Merging of two LAMP1/SirLyso-positive mature lysosomes or one LAMP1/SirLyso-positive mature lysosome and one LAMP1-positive immature lysosomes were considered as fusion events while splitting of two LAMP1/SirLyso-positive mature lysosomes or splitting of LAMP1-positive lysosomes from LAMP1/SirLyso-positive mature lysosomes were considered as fission events. The number of fusion and fission events on all LAMP1/SirLyso-positive lysosomes (±1588 in control, ±643 in RTN4/DP1 KD and ±717 in P180 KD) from the live soma images of 10–14 neurons per condition were counted manually for 301 frames (1 frame/sec) by three

independent observers. The counts were averaged and plotted per soma (Figs. 4k, m, 6j) or per lysosome by dividing the number of fusion or fission events by the total number of LAMP1/SirLyso-positive lysosomes per soma (Fig. 4l, n; Supplementary Fig. 6g, h). The fusion/fission ratio was calculated by dividing the total number of fusion events per soma by the total number of fission events per soma (Fig. 4o and Supplementary Fig. 6i). We used the same control neurons for comparisons with RTN4/DP1 KD neurons or P180 KD neurons in all analyses as the experiments were performed together. Somatic lysosome motility was quantified from the same image sequences used for the quantification of the total number of lysosomes per soma. To track both motile and immotile LAMP-positive lysosomes with more precision during the 5-min recording, we counted all immotile lysosomes and inferred the motile population by subtracting the immotile lysosomes from the total number of lysosomes. Lysosomes undergoing only short displacement because of remodeling (e.g. lysosomes undergoing fusion and fissions) were considered immobile. Lysosome motility was expressed as a percentage of the total number of lysosomes in Supplementary Figs. 3a, b, 6a.

*Quantification of immunofluorescence intensity for Strep.* All images were taken with the same settings for the strep channels including laser power, exposure and gain, and pixel intensities were kept below saturation. Quantification of the intensity of Strep signal was performed using ImageJ. z-projections of each image were generated using the average intensity and a ROI was manually drawn around the neuronal soma. Mean intensities from 16-bit images for one channel corresponding to Strep signal in the selected area was measured using ImageJ. Intensities were averaged over multiple cells and normalized to the average intensity in control cells (Fig. 5c; Supplementary Fig. 6k, m). Quantification of the intensity of Strep signal was also performed for non-pre-axonal and pre-axonal regions using ImageJ. A 5–7 µm long line was drawn from the beginning of the AIS (labeled with TRIM46 antibody) towards the soma to define the pre-axonal region. A ROI was manually drawn around this region of the soma and defined the border of pre-axonal region. Another ROI was manually drawn excluding this area of the soma defining the border of the non-pre-axonal region (Fig. 5i). The intensity measurements and normalization were performed as explained above (Fig. 5j, k).

*Quantification of SirLyso intensity.* All images were acquired with the same settings for the SirLyso channel, including laser power, exposure and gain, and pixel intensities were kept below saturation. Quantification of the intensity of the SirLyso signal was performed using ImageJ. A ROI was manually drawn around the neuronal soma and mean intensities from 16-bit images for the channel corresponding to the SirLyso signal in selected areas was measured using ImageJ. Intensities were averaged over multiple cells and normalized to the average intensity in control cells (Supplementary Fig. 3j).

**Statistical analysis.** Data processing and statistical analysis were performed using Excel and GraphPad Prism (GraphPad Software). Unpaired and paired *t*-test, Kruskal–Wallis test followed by a Dunn's multiple comparison test, Mann–Whitney *U*, Wilcoxon test, one-way ANOVA test followed by Tukey's or Sidak's multiple comparisons test, were performed for statistical analysis and are indicated in Figure legends. Significance as determined as followings: ns- not significant, *$p$ < 0.05 **$p$ < 0.01 and ***$p$ < 0.001. The assumption of data normality was checked using D'Agostino–Pearson omnibus test.

**Reporting summary.** Further information on research design is available in the Nature Research Reporting Summary linked to this article.

## Data availability
The proteomics dataset that was analyzed in this study was deposited for a previous publication (Farias et al., 2019). This dataset was deposited onto the ProteomeXchange Consortium via the PRIDE partner repository with the dataset identifier PXD012264. This is publicly accessible at. All data that support the findings of this study are included in the manuscript or are available from the authors upon reasonable request. Raw data and uncropped Western blots are available in the Source Data file. Source data are provided with this paper.

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

## Acknowledgements

We thank Dr. Juan Bonifacino (NIH) for sharing the GFP-KIF5A-rigor and mCh-KIF5A-Strep constructs. We acknowledge C. de Heus and T. Veenendaal of the Cell Microscopy Center UMC Utrecht for their valuable assistance with CLEM experiments. We thank Dr. Anna Akhmanova for critically reading the manuscript. This work was supported by the Netherlands Organization for Scientific Research (NWO) through a VIDI grant (016.VIDI.189.019) to G.G.F., a KLEIN grant (OCENW.KLEIN.236) to G.G. F. and J.K., a VENI grant (VI.VENI.202.113) to M.K., a NWO Roadmap on Netherlands Electron Microscopy infrastructure NEMI (project 184.034.014) to J.K., a ZonMW-TOP grant (91216006) to J.K., and a ZonMW-TOP (91217002) grant to L.C.K. Additional support came from the European Research Council (ERC-StG 950617 to G.G.F., ERC-CoG 819219 to L.C.K), Alzheimer Nederland (WE. 15045 to C.C.H., WE.03-2019-10 to J. K.), and the Deutsche Forschungs Gemeinschaft (DFG FOR2625 to J.K.).

## Author contributions

N.Ö. designed and performed experiments, analyzed data, and wrote the manuscript; M. K. designed and performed experiments, analyzed data, and wrote the manuscript; I.v.S. performed experiments related to enlarged lysosome after ER tubule disruption; A.v.H. performed experiments related to P180; D.J. performed expansion microscopy experiments under guidance of L.C.K.; N.L., designed and performed correlative light and electron microscopy experiments; J.K., and L.C.K. discussed data, provided feedback and edited the manuscript; C.C.H. proposed experiments, discussed data and provided feedback and edited the manuscript; G.G.F. designed and performed experiments, analyzed data, supervised the research, coordinated the study, and wrote the manuscript.

## Competing interests

The authors declare no competing interests.
