## [Peer Review File · Nature Communications]

REVIEWER COMMENTS

Reviewer #1 (Remarks to the Author):

The manuscript by Özkan et al. investigates ER – lysosome contacts and their role in regulating axonal lysosome translocation. Using a heterodimerization system to control ER tubule distribution, the authors show that somatic ER tubules are required for lysosomal transport into the axon. They further identify that ER tubules regulate axonal lysosomal availability by modulating lysosomal size via lysosome homo-fission and that contacts between ER tubules and lysosomes are enriched in the pre-axonal region. Finally, they find that the ER protein P180, which they previously identified as important for ER-microtubule interaction, modulates lysosomal translocation into the axon independent of ER-lysosome contact tethering.

Considering the emerging role inter-organelle contacts in neuronal function, this study is interesting, and the experiments are logical and well-performed. However, the finding that ER-lysosome contacts regulate lysosomal fission and axonal lysosomal availability is a modest advance in light of previous findings from this group and others (Farías et al., 2017; Farías et al., 2019; Rowland et al., 2014; Allison et al., 2017). In addition, a number of controls are missing, and the emphasis on P180, which has been suggested to have multiple functions in addition to stabilizing ER-microtubule interactions, warrants further investigation particularly as a mechanistic link between P180 and ER-lysosome contacts is not well-explored. Addressing the technical and conceptual concerns below will considerably strengthen the manuscript.

Major Points

- 1) The authors report that ER morphology regulates axonal lysosomal availability by assessing lysosomal distribution in somatodendritic and axonal regions upon knockdown of ER shaping proteins. To strengthen these claims, please quantify the knockdown efficiency of RTN4/DP1 or CLIMP63.
- 2) Similarly, how do the authors discriminate between axons and dendrites in their quantification? It would be important to include additional marker proteins (e.g. MAP2 for dendrites) to more accurately characterize neurites for quantification of lysosomal availability in these different compartments.
- 3) Knockdown of RTN4/DP1 appears to affect both anterograde and retrograde Lamp1 transport (Figure 1F). Are ER tubules important for lysosome transport from the axon back to the soma as well? Alternatively, does knockdown of RTN4/DP1 affect lysosomal motility globally? This may be assessed by comparing somatic lysosomal motility between wild-type and RTN4/DP1 knockdown cells.
- 4) The authors show that local ER tubule disruption or depletion of somatic ER tubules cause enlarged, less motile lysosomes in the soma. Because lysosomal enlargement and decreased motility are also characteristic of lysosomal cell death, how are the authors certain that the observed phenotypes are because of altered ER morphology rather than overall decrease in cell viability? Please address by assessing whether RTN4/DP1 knockdown and SBP-RTN4A/KIF5A-Strep expression affect cell viability.

5) In Figure 2E, the authors show that expression of KIFC1-Strep does not alter Lamp1 transport. However, in Video S4, KIFC1-Strep seemingly increases lysosomal motility. How do the authors reconcile increased lysosomal motility with unaffected lysosomal transport? Are these findings compartment-specific? Please quantify lysosomal motility in the KIFC1-Strep condition.

6) The authors use APEX labeling with protrudin and Rab7 to identify ER-lysosome contacts in the soma. However, Rab7 recruitment to the lysosome is regulated by its GTPase activity and thus, there is also a cytosolic pool of Rab7. How do the authors exclude the possibility of cytosolic Rab7 interacting with protrudin to generate APEX labeling (i.e. APEX labeling that does not occur at ER-lysosome contacts)?

7) The authors show example images of ER-lysosome contacts enriched in the pre-axonal region. To further support this conclusion, it is necessary to quantify ER-lysosome contacts in pre-axonal and non-pre-axonal regions either via Strep568 intensity or other methods.

8) The authors suggest that ER-lysosome contacts in the pre-axonal region are important for lysosomal fission and translocation into the axon. Are ER tubules present at sites of lysosomal fission in the pre-axonal region? Because perturbations on ER tubules affect lysosomal fission, does the percentage of lysosomal fission events marked by ER tubules differ in control vs. ER tubule-disrupted cells?

9) Knockdown of P180 appears to have a prominent effect on lysosomal size, motility and translocation, but the effects appear to be independent of ER-lysosome contact formation. Given that the lysosomal phenotypes of P180 knockdown and ER tubule-lysosome contact disruption are similar, does P180 knockdown affect other aspects of ER-lysosome contact dynamics such as tethering duration, etc?

10) Are P180 knockdown-induced defects in transport from the pre-axonal region into the axon specific for lysosomes or is transport of other organelles (i.e. other endolysosomal vesicles, mitochondria, etc.) also affected?

11) Quantification for several panels (e.g. Figures 1F, 3E-I, 4K-L, 6B) appears to be derived from analysis of n = 8 neurons. While the results appear robust, given that these analyses were conducted on transiently transfected primary neurons (with likely low transfection efficiency), it would be advantageous to increase the n value. What is the transfection efficiency of these neurons? Please also include additional replicates in the analyses.

Minor Points

1) Please quantify P180 levels to verify knockdown efficiency (Figure 6).

2) In addition to microtubule stabilization, P180 has been reported to play a role in ribosome-independent mRNA localization to the ER (Cui et al., 2012). Please include discussion of other functions of P180 and how they may/may not interface with ER-lysosome contact dynamics and function.

3) The labeling in Figure 3C lower two panels should be corrected (i.e. change "SBP-RTN4A + KIF5A-FRB" to "SBP-RTN4A + KIF5A-Strep").

Reviewer #2 (Remarks to the Author):

How neuronal cells maintain an organized anterograde and retrograde flow of organelles within their axonal and dendritic compartments is a key but poorly understood problem. The manuscript by Ozkan et al proposes a role for tubular ER in mediating the transport of lysosomes, which are key mediators of cellular catabolism and metabolic signaling, within the axon. Building on recent work showing a key role for the ER in facilitating lysosomal fusion/fission, the authors provide evidence that physical contact between ER and lysosomes at the pre-axonal region promotes loading of a specialized kinesin onto lysosomes, resulting in their fission and movement down the axon. The final model proposes that the ER may thus act as a gatekeeper that regulates the number and size of lysosomes that are allowed into the axon.

This is a very interesting and well executed manuscript that is supported by high quality biochemical and imaging data.

A few suggestions for improvement are listed below.

1. In Fig. 1 and subsequent figures, the authors show that knocking down ER-organizing factors such as RTN4 and DP1 causes clear changes in the distribution of lysosomes and their morphology. Given the role of lysosomes in supporting mTORC1 signaling, the authors should check the status of canonical mTORC1 substrates (i.e. S6K, 4EBP1, TFE3) upon depletion of these factors.
2. Connected to the previous point, do these alterations of ER morphology and the resulting impairment of lysosomal positioning impact autophagy initiation, progression or termination? This could be ascertained using markers such as the green-red-LC3, as well as staining for Atg13 puncta.
3. Although the evidence supporting the requirement for P180 in lysosomal loading onto microtubules is strong, the exact relationship between P180 and kinesin-1 remains unclear. Recently, the multisubunit BORC complex was shown to promote kinesin loading onto lysosomes via the small GTPase Arl8 (PMID: 25898167). Does P180 interact with BORC or Arl8?
4. Given the role of ER in mitochondrial homeostasis, and the feedback effects that mitochondrial status exerts on lysosomes (i.e. via AMPK) it would be good to test mitochondrial function upon depletion of the ER-organizing factors, including membrane potential and redox status.

Reviewer #3 (Remarks to the Author):

In their manuscript titled "ER – lysosome contacts at a pre-axonal region regulate axonal lysosome availability", the authors explore a connection between lysosome transport in the axon and ER tubules in the soma of cultured neurons. They broadly conclude that somatic ER tubules regulate lysosome size and axonal translocation by promoting lysosome homo-fission, and suggest a model wherein ER tubule – lysosome contacts at the somatic pre-axonal region promote kinesin-1-powered lysosome fission and subsequent axonal translocation. These claims are novel and would be of interest to cell- and neuro- biologists. Unfortunately, I did not find that the experimental data supports either of these broad conclusions.

Major comments

ER shape regulates lysosome availability in the axon

1. The authors conclude from the experiments summarized in figure 1 that they show "ER tubules play a critical role in regulating lysosome translocation from the soma into the axon." This conclusion is not supported by the data. The authors do not show that the phenotype of RTN4 + DP1 KD is exclusively or even mainly a reduction in ER tubules. Yet, they draw a direct link between the KD and the morphology of the ER. While this may be one of the phenotypes supported by the literature, interfering with ER morphology would have a pleiotropic effect on the cell. In particular membrane proteins like LAMP1 and on the homeostasis of the entire endomembrane system, of which lysosomes are only one branch. How did the authors exclude an indirect effect that would explain the results?

2. I am not convinced the polarity index calculation can be taken as a surrogate for lysosome distribution (figure 1). The mean fluorescence intensity would correlate with the average size of the lysosomes as well as their distribution. Consider a scenario where the amount of lysosomes in the dendrite is much larger but they are all dimmer than those of in the axon? How do the authors assure this is not the case? The authors should show the distribution of fluorescence intensities in axonal and dendritic lysosomes. In figure 3 the authors actually show phenotypes associated with lysosome size using the same shRNA.

Somatic, but not axonal, ER tubules promote lysosome translocation into the axon

3. How do the authors exclude the possibility that when they induce axonal transport of ER tubules (Figure 2; "+ Strep-KIF5A") the phenotype they observe is not due to lack of space or lack of available motors for the lysosomes? Basically, why do they think it is anything beyond a simple traffic jam. Local ER tubule disruption causes enlarged and less motile mature lysosomes in the soma

4. In Figure 3 – Even if the author's assumption that RTN4 + DP1 KD leads to a reduction in tubules. Would not the reduction be everywhere and not just in the soma? Can the authors offer some explanation why the phenotype of the KD is similar to the somatic reduction alone? The images actually suggest the lysosomes are larger under shRNA conditions than under somatic reduction alone.

5. I did not understand how the authors conclude that on the one hand "ER tubule disruption did not affect lysosome activity" and on the other hand that "mature lysosome population (LAMP1 / SirLyso positive) under ER tubule disruption was reduced to 33% ". Mature lysosomes and lysosome activity were defined by the same method (SirLyso). I would argue that lysosome activity is the cumulative activity of the lysosomes in the cell and not the individual activity of each lysosome. Even if I was to accept the later, the authors only show that some activity is measurable without truly quantifying SirLyso or MagicRed fluorescence intensity or showing that cathepsin translation and transcription are unaltered under these experimental conditions. The authors should offer some validation of their experimental model in these regards, preferably using standard biochemistry.

6. Given that the authors disrupted ER morphology how can they be sure the LAMP1 signal still defines lysosomes? In particular, in light of the fact that LAMP1 and SirLyso don't fully overlap under RTN4 + DP1 KD. I cannot judge it for the somatic depletion (figure 3C) because the data is not shown. The authors casually state that "lysosome activity was often observed compartmentalized" but offer no evidence that these can be defined as lysosomes at all and not an artificial intermediate. CLEM data could have been potentially used to address this directly.

7. I also have major concerns regarding the CLEM experiments. I did not understand why CLEM is used to measure the diameter of micron sized objects? Can the authors show what is the difference if the diameter is measured from the fluorescence data? The advantage being that FM data does not undergo shrinkage. The authors should also measure the diameter of lysosomes in the control. I did not understand where 400nm came from. I also did not find any statistics. How many times the

experiment was repeated, how many cells were analyzed etc. The authors state that “The compartmentalized fluorescence SirLyso signal corresponded to the areas with intraluminal vesicles “. I wonder if the authors have enough resolution and correlation precision to support this statement. The correlation was done manually on resliced images so how can they be sure. Take for example lysosome number 2. I see no correspondence between the SirLyso and intraluminal vesicles.

8. “Since RTN4/DP1 knockdown also reduced the total number of lysosomes (Figure 3E)”, how can the authors conclude anything about their distribution (figure 1) without normalizing the data on the overall number?

ER tubules regulate lysosome homo-fission

9. How can the conclusion from these results even remotely suggest anything about ER tubule – lysosome contacts. The authors should really be more precise throughout the manuscript and avoid jumping to conclusions.

Minor comments

1. The paper is well written but can improved with less Jargon and by explaining the rationale of the experiments. Otherwise it may not be suitable to a broad audience.

Examples:

a. day-in-vitro 7 (DIV7) – would only be obvious to someone in the field.

b. “Quantification of the polarity index” why was it calculated, what was the rationale?

2. Unless citations are highly limited Fiji/ImageJ and all plugins used should be acknowledged by citing the associated publication.

3. In ShRNA experiments the details of the control are not provided. The standard control for ShRNA is a non-targeting shRNA sequence. Minimally, an empty vector control would be acceptable.

4. It should be clarified in the text that Figure 1 A and C and figure 3 A and B are the same experiment. The only difference is in what was quantified.

5. The authors may consider always having an ER tubule fluorescent reporter as a co-transfection marker for shRNA experiments.

6. The statement “lysosomes of remarkable consistent size and shape “ is imprecise and should be removed. The lysosomes in question are simply spherical and their size distribution is actually larger than the overall size of the WT lysosomes (400nm), so how can it be defined as consistent.

Moreover, the number of measurements done seems extremely small and ignores the majority of LAMP1 fluorescent puncta.

7. The number of cells quantified and the number repeats should be stated for each experiment.

8. Rab7 is a classical late endosome marker and not a lysosome marker, unless it co-localizes with other lysosomal markers like LAMP1. RAB7 overexpression has been recently shown to induce the formation of triple contact sites between ER late endosomes and mitochondria (see <https://doi.org/10.1073/pnas.1913509116> and <https://www.nature.com/articles/s41467-020-17451-7>).

9. Can the authors not use CLEM to demonstrate the existence of contact sites at the pre-axonal region?

Reviewer #4 (Remarks to the Author):

In nonneuronal cells, it is known that ER-lysosome contact sites are required to enable lysosome

homo-fission, and kinesin is involved in lysosome fission. ER structure is highly dynamic; it undergoes remodeling in the order of seconds. It is not clear how the local ER organization may control the LE/lysosome size and function. Here the authors show that in rat hippocampal neurons, knockdown of the ER tethering proteins VAPs does not produce disruption in ER lysosome contact, suggesting that redundancy may exist in membrane tethering events that involve ER. Instead, knockdown of proteins that control the ER shape, RTN4, and DPI, causes ER tubule disruption, causes disruption in ER-lysosome contact sites, and interferes with lysosomal homo-fission. They also show that in neurons, the ER-protein P180 binds microtubules to promote kinesin-1 dependent lysosome fission.

This work emphasizes the importance of ER shape in controlling interactions between the ER membrane and the endo/lysosome membrane. The experiments are extensive and carefully performed. The work is novel and interesting.

Comments-

1. RESULT- Please report efficiencies in individual KD experiments. KD VAPs do not produce disruption in ER lysosome contact. How was the contact sites monitored? Could this be due to a lack of efficiency in VAP KD?
2. Please provide evidence that ER shape is changed after KDs of RTN4 and DPI.
3. ABSTRACT. I suggest that the words " KD proteins that control the ER shape, RTN4 and DPI, causes ER tubule disruption and causes disruption in ER-lysosome contact" be included in the abstract.
4. INTRODUCTION-It needs to be revised; sentences that describe the results should be deleted.

Point-by-point response to the reviewers:

***** Reviewer #1 *****

Considering the emerging role inter-organelle contacts in neuronal function, this study is interesting, and the experiments are logical and well-performed. However, the finding that ER-lysosome contacts regulate lysosomal fission and axonal lysosomal availability is a modest advance in light of previous findings from this group and others (Farías et al., 2017; Farías et al., 2019; Rowland et al., 2014; Allison et al., 2017). In addition, a number of controls are missing, and the emphasis on P180, which has been suggested to have multiple functions in addition to stabilizing ER-microtubule interactions, warrants further investigation particularly as a mechanistic link between P180 and ER-lysosome contacts is not well-explored. Addressing the technical and conceptual concerns below will considerably strengthen the manuscript.

R: We thank the reviewer for his/her positive assessment of our study and the constructive criticism on our manuscript. We believe that our new data further strengthens the main conclusion of the manuscript. We believe our findings are novel, as this is the first time that it is shown how ER organization (as ER tubules and cisternae) regulate contact formation between the ER and lysosomes. A multi-steps mechanism is proposed, in which ER tubule – lysosome – MT contacts at a pre-axonal region promotes lysosome fission followed by lysosome translocation into the axon. We have added our final model in Figure 7E. Below, you will find our responses to the specific concerns.

Major Points

1) The authors report that ER morphology regulates axonal lysosomal availability by assessing lysosomal distribution in somatodendritic and axonal regions upon knockdown of ER shaping proteins. To strengthen these claims, please quantify the knockdown efficiency of RTN4/DP1 or CLIMP63.

R: We agree with the reviewer that validation of the knockdown efficiency is key to strengthen our claims. We have previously validated the knockdown efficiency for rat shRNA-RTN4 and rat shRNA-DP1 in our lab (Farías et al., Neuron, 2019, in Supplemental Figure S3). Rat shRNA-CLIMP63 has been previously validated by the Ehlers lab (Cui-Wang et al., Cell, 2012, in Figures 6F-G and 7G) and the same sequence was used here, as well as in our previous paper (Farías et al., Neuron, 2019). We have stated this more clearly in the Methods section. In addition, we have added the quantitative proteomics results of RTN4-KD and DP1-KD neurons to Figures S3K and S4E (proteomics data from Farías et al., 2019) together with the analysis of lysosomal and autophagy-related proteins.

As a proof of reorganization of the ER upon RTN4 plus DP1 knockdown, we have included additional experiments. We have applied the newly developed Ten-fold Robust Expansion microscopy (TReX) (Damstra et al., 2021; BioRxiv) to examine the nanoscale structure of the ER in control neurons and impaired ER tubule formation upon knockdown of RTN4 plus DP1 (Figure S1; Video S1). We also included live cell imaging to show the dynamics of ER tubules at the pre-axonal and proximal axon regions in control cells and ER tubule disruption and absence of the ER at the pre-axonal and proximal axon regions upon knockdown of RTN4 plus DP1 (Video S2).

2) Similarly, how do the authors discriminate between axons and dendrites in their quantification? It would be important to include additional marker proteins (e.g. MAP2 for dendrites) to more accurately characterize neurites for quantification of lysosomal availability in these different compartments.

R: All the images that were used for our polarity index quantifications come from neurons with morphologically well-defined axons and dendrites in the in vitro developmental stages we study (DIV6-DIV8). We routinely perform this type of analysis for assessing the polarized sorting of proteins and organelles, in which we identify the axon and dendrites based on the length of axons (defined as at least three times longer than dendrites) using GFP, mCherry or BFP empty vectors as a fill, and/or using very well characterized markers for the axon initial segment such as TRIM46 or Neurofascin (NF; explained in Figure legends and Methods) which are absent in dendrites (Farías et al., Cell Reports, 2015; Van Beuningen et al., Neuron, 2015; Farías et al., PNAS, 2017, Farías et al., Neuron, 2019). In addition, our polarity index quantifications were performed independently by at least two researchers, obtaining similar results. We have included more detailed information on how axons and dendrites were identified in this study in the Methods section.

3) Knockdown of RTN4/DP1 appears to affect both anterograde and retrograde Lamp1 transport (Figure 1F). Are ER tubules important for lysosome transport from the axon back to the soma as well? Alternatively, does knockdown of RTN4/DP1 affect lysosomal motility globally? This may be assessed by comparing somatic lysosomal motility between wild-type and RTN4/DP1 knockdown cells.

R: We appreciate this comment from the reviewer regarding LAMP1 motility. We show in Figure 1F that knockdown of RTN4/DP1 reduces both anterograde lysosomal transport into the axon and retrograde transport back to the soma. The number of stationary lysosomes is not increased, but the total number of lysosomes in the axon is reduced (anterograde + retrograde + stationary). We do not observe more lysosomes pausing or stopping along the axon, suggesting that there is no defect in the axonal displacement of the fewer lysosomes that do enter into the axon. This suggests that the defect in lysosome distribution and dynamics in the axon is likely a consequence of less lysosomes entering into the axon in the first place, which subsequently impacts their retrograde transport back to the soma. We confirmed this in Figure 2 by analyzing the local role of the ER in the soma by controlling ER repositioning from soma into the axon, and in Figure 3 by analyzing the defects in somatic lysosomes. As we show in Figure 3, related videos S5 and S6, enlarged and less motile lysosomes were found in the soma upon both RTN4/DP1 KD and ER repositioning from the soma into the axon. We have toned down our conclusion for Figure 1 and added quantifications for the total number of lysosomes along a segment of the axon in Figure 1F. Prompted by the reviewer's suggestion, we have quantified lysosome motility in the soma not only for knockdown of RTN4+DP1, but also for repositioning of somatic ER tubules into the axon, and for knockdown of the MT- and kinesin-1 binding protein P180. The motile pool of somatic lysosomes was reduced in all these three conditions compared to control, consistent with our conclusion that somatic ER tubule – lysosome contacts promote lysosome fission and translocation of lysosomes into the axon. These results related to Figure 3 are now shown in Figures S3A, S3B and S6A.

4) The authors show that local ER tubule disruption or depletion of somatic ER tubules cause enlarged, less motile lysosomes in the soma. Because lysosomal enlargement and decreased motility are also characteristic of lysosomal cell death, how are the authors certain that the observed phenotypes are because of altered ER morphology rather than overall decrease in cell viability? Please address by assessing whether RTN4/DP1 knockdown and SBP-RTN4A/KIF5A-Strep expression affect cell viability.

R: It is indeed a very important point to address if the observed phenotype is specific for altered ER morphology and is not due to an overall decrease in cell viability. None of the analyzed neurons showed any morphological characteristics of dying cells, such as fragmentation of dendrites or the axon, as shown by expressing a cytosolic fluorescent protein used such as GFP, mCherry or BFP as a fill.

In addition to this, we have now performed nuclear DAPI staining to assess if these conditions cause an increase in intensity/condensation of DAPI, which is an indicator for apoptosis. We have counted the number of transfected cells with a normal DAPI versus an increased or aberrant DAPI staining. This showed that neither RTN4/DP1 knockdown nor SBP-RTN4/KIF5A-Strep expression had a significant effect on cell viability compared to transfected (pSuper or SBP-RTN4 only) control cells (revised Figure S2B). We have also assessed the distribution of mitochondria in neurons, which are very susceptible to cell damage. The distribution of mitochondria in the axon, as quantified by polarity index calculation, was not reduced by RTN4/DP1 knockdown or by SBP-RTN4/KIF5A-Strep expression (Figures S2C, S2D, S2G and S2H). Finally, we also assessed the distribution of Rab3- and Rab11-positive vesicles for knockdown of RTN4 plus DP1, and they also showed normal distribution compared to control (Figure S2E, S2F, S2I and S2J). Together, this suggest that the reduced axonal distribution of lysosomes and somatic lysosomal enlargement we observe after local ER tubule disruption are not caused by a decrease in cell viability or because a pleiotropic effect. All these results are now described in the revised manuscript.

5) In Figure 2E, the authors show that expression of KIFC1-Strep does not alter Lamp1 transport. However, in Video S4, KIFC1-Strep seemingly increases lysosomal motility. How do the authors reconcile increased lysosomal motility with unaffected lysosomal transport? Are these findings compartment-specific? Please quantify lysosomal motility in the KIFC1-Strep condition.

R: We thank the reviewer for pointing out this possible contradiction in our results. Indeed, our quantification shows unaffected axonal lysosomal transport when pulling the ER into the soma (Figure 2E) while the motility of lysosomes in the soma shown in Video S4 seems to be increased by eye. We have now quantified lysosome motility in somas of the KIFC1-Strep condition and compared this with the control condition. This quantification shows no significant increase in the motility of somatic lysosomes. We have added this quantification to our manuscript in Figure S3B and we have replaced the original Video S4 to show more representative movies, now shown as Video S6.

6) The authors use APEX labeling with protrudin and Rab7 to identify ER-lysosome contacts in the soma. However, Rab7 recruitment to the lysosome is regulated by its GTPase activity and thus, there is also a cytosolic pool of Rab7. How do the authors exclude the possibility of cytosolic Rab7 interacting with protrudin to generate APEX labeling (i.e. APEX labeling that does not occur at ER-lysosome contacts)?

R: We appreciate the reviewer for raising this concern. To determine whether APEX (streptavidin) labeling occurs specifically at ER-lysosome contacts, we designed a Split-EX-HA-Rab7 T22N mutant that was previously shown to remain cytosolic (Bucci et al. Mol. Biol. Cell, 1999) and then performed our split-APEX assay. We observed a clearly cytosolic distribution of our Split-EX-HA-Rab7 mutant, as well as a diffuse cytosolic Streptavidin signal, which was not enriched or colocalizing with protrudin, in contrast to wildtype Rab7 (Figure 5B and 5D). Further supporting that our APEX labeling occurs at ER-lysosome contact sites, we found that VAPB overexpression significantly increases APEX labeling (Figure S6L and S6M, and further discussed in point 9). However, we cannot completely exclude the possibility of APEX labeling arising from an interaction between

cytosolic Rab7 and protrudin with this technique. We now state in the manuscript that these results suggest that interactions occur mainly when the labeled lysosomal protein is associated to the lysosomal membrane (Figure 5D).

7) The authors show example images of ER-lysosome contacts enriched in the pre-axonal region. To further support this conclusion, it is necessary to quantify ER-lysosome contacts in pre-axonal and non-pre-axonal regions either via Strep568 intensity or other methods.

R: We appreciate this comment and agree that this quantification provides further support for our model. We therefore quantified Streptavidin intensity in both the pre-axonal region and the rest of the soma. This quantification revealed that our APEX (streptavidin) labelling was enriched at the pre-axonal region, which was defined by a 5- μ m line from the beginning of the AIS (Farías et al., Cell Reports, 2015) to the cell body, compared to the rest of the soma (Figures 5I-K). This new result is described in the revised manuscript.

8) The authors suggest that ER-lysosome contacts in the pre-axonal region are important for lysosomal fission and translocation into the axon. Are ER tubules present at sites of lysosomal fission in the pre-axonal region? Because perturbations on ER tubules affect lysosomal fission, does the percentage of lysosomal fission events marked by ER tubules differ in control vs. ER tubule-disrupted cells?

R: We agree with the reviewer that it is important to visualize ER tubules during lysosomal fission at the pre-axonal region and examine the effect of ER tubule disruption in lysosomal fission. This proposed experiment, showing all our observations together, is extremely difficult to achieve, because i) the thin and small structure of the pre-axonal region compared to the soma, ii) the very dense packed material in a narrow space and iii) difference in thickness between the soma and proximal axon that make it difficult to get all the occurring events in focus at the pre-axonal region during fast live cell imaging for several markers. Having said that, we have nonetheless accomplished to show ER tubules and lysosome contacts in live neurons at the pre-axonal region, as well as lysosome fission and translocation events occurring at these contact sites in control neurons. We also compared ER tubule-disrupted and P180-KD neurons with control neurons. All these results are now shown as a new main Figure 7 and are described in the revised manuscript.

In detail, we have applied two different approaches in order to visualize the presence of ER tubules at the site of fission and translocation of lysosomes in the pre-axonal region. As a first approach, we labelled the ER with the general ER marker Sec61 β -GFP, lysosomes with SirLyso dye and the AIS with TRIM46-BFP and performed live cell imaging (Figure 7A and Video S11). As a second approach, we introduced the reversible dimerization-dependent fluorescent protein (ddFP) domains GB and RA (Ding et al., Nat Methods, 2015), which have recently been used to illuminate ER-P body contact in live cells (Lee et al., Science, 2020), into neurons in order to visualize ER-lysosome contacts during live cell imaging (Figure 7B). We labelled lysosomes with the lysosomal marker LAMP1-GFP, ER-lysosome contacts by using GB-Rab7 and RA-Protrudin and labeled the AIS with NF-647 followed by live cell imaging.

With these two approaches we were able to visualize lysosome fission and translocation events associated to contact sites at the pre-axonal region. In addition, we show with the GB/RA contact assay that ER tubule depletion causes reduced contacts, and lysosome fissions and translocation events were less often observed at the pre-axonal region. As explained above, there are many technical challenges in performing these types of experiments. We have obtained several videos showing the same results consistently, represented in the images

and videos added as Figure 7A, 7C, 7D and Video S11. However, it would be impossible to quantify the number of all fissions occurring in contact with the ER with precision in the small, crowded and not always in-focus pre-axonal region, and to quantitatively compare this with knockdown conditions.

In addition, we now show live cell imaging for the general ER marker Sec61 β for control and RTN4 plus DP1 knockdown neurons in new Video S2. In control conditions, dynamic ER tubules are present in the pre-axonal region, while knockdown of RTN4+DP1 causes ER tubule to ER cisternae conversion and retraction of ER membranes into the soma, which were less dynamic and absent from the pre-axonal region (consistent with our previous work in Farías et al., Neuron, 2019). We believe that adding all these new results strengthens our model now graphically represented as Figure 7E.

9) Knockdown of P180 appears to have a prominent effect on lysosomal size, motility and translocation, but the effects appear to be independent of ER-lysosome contact formation. Given that the lysosomal phenotypes of P180 knockdown and ER tubule-lysosome contact disruption are similar, does P180 knockdown affect other aspects of ER-lysosome contact dynamics such as tethering duration, etc?

R: It is indeed possible that P180 knockdown affects other aspects of ER-lysosome contact formation and we were initially surprised that P180 knockdown did not affect ER-lysosome contacts. However, previous studies have shown that depletion of several tethering proteins that are involved in ER-lysosome contact formation (VAPB, ORP1L and STARD3) does not significantly decrease ER-lysosome contacts (discussed in Lee and Blackstone., 2020; Fowler et al., 2019). Therefore, we overexpressed the classical tethering protein VAPB and P180 and quantified Streptavidin labelling using our Split-APEX assay to further investigate the involvement of P180 in ER-lysosome contact formation. As expected, VAPB overexpression produced a significant increase in APEX Strep labelling, while P180 overexpression did not increase Strep signal compared to control neurons (Figures S6L and S6M). To further determine whether P180 knockdown affects tethering duration we introduced the reversible GB-RA contact assay (Lee et al., Science, 2020) as explained above in point 8. We observed that in control neurons ER – lysosomes contacts were visible during our 120-second live cell imaging, consistent with previous works showing these contacts are very stable (Friedman et al., Mol. Biol. Cell, 2013; Guo et al., Cell, 2018). Similarly, these contacts remained visible during same time period for P180 knockdown, but fission and translocation were affected at the pre-axonal region (Figure 7C, 7D and Video S11). Because both the MT- and Kinesin-1 binding domains of P180 are required for lysosome translocation into the axon (Figure 6G-I), we propose that P180 may contribute to a final step in contact – MT stabilization for subsequent kinesin-1-powered lysosome fission and translocation into the axon. We have added these results in the mentioned Figures and described them in the revised manuscript.

10) Are P180 knockdown-induced defects in transport from the pre-axonal region into the axon specific for lysosomes or is transport of other organelles (i.e. other endolysosomal vesicles, mitochondria, etc.) also affected?

R: We appreciate this comment from reviewer. We found that the MT-binding domain of P180 is important for MT-ER co-stabilization (Farías et al., 2019) and both MT- and kinesin-1 binding domains are required for axonal transport of lysosomes (Figure 6H). It is indeed important to clarify whether the effect of P180 knockdown on transport into the axon is unique to lysosomes or whether it is required for general transport of other organelles as well. Knockdown of P180 causes impaired axonal distribution of lysosomes (as shown in original Fig S2A-B, now Figure S5A-B). We have now assessed the distribution of mitochondria that, like lysosomes, also require kinesin-1 for axonal transport into the axon (van Spronsen et al., Neuron, 2013; Farías et al., PNAS, 2017). In

addition, we have also analyzed the distribution of the mainly axonal Rab3-positive vesicles and the mainly somatodendritic Rab11-positive endosomes to assess if P180 knockdown affects the transport of other organelles into the axon. We found that P180 knockdown does not significantly affect the distribution of these organelles along the axon. We have included representative images and quantifications for mitochondria, Rab3- and Rab11-positive vesicles after P180 knockdown in the revised manuscript (new Figures S2C, S2E, S2F, S2G, S2I and S2J).

11) Quantification for several panels (e.g. Figures 1F, 3E-I, 4K-L, 6B) appears to be derived from analysis of n = 8 neurons. While the results appear robust, given that these analyses were conducted on transiently transfected primary neurons (with likely low transfection efficiency), it would be advantageous to increase the n value. What is the transfection efficiency of these neurons? Please also include additional replicates in the analyses.

R: Transfection efficiency in primary hippocampal neurons is indeed relatively low at approximately 7-10%. However, we performed this analysis from live-cell imaging data that were obtained from at least two independent experiments and quantifications were performed by at least two different researchers, obtaining similar results. Live cell imaging and quantification in neurons is difficult and time-consuming. To provide even more robust results for live cell imaging and quantification related to axonal transport of lysosomes (Figure 1F, 2F-E, 6B) we now increased sample size to 16-20 neurons per condition, from 3-4 independent experiments. For quantifying lysosome size, fission and fusion events we also increased sample size to 13-14 neurons per condition from 3-4 independent experiments (Figures 3E-I; Figures 4K-O and Figures S6B-I). It may be good to note that we obtained this data by observing and quantifying a very large numbers of lysosomes (SirLyso+ lysosomes: 1588 in control, 643 in RTN4/DP1 KD and 717 in P180KD conditions). Importantly, these additional quantifications showed similar results. We have included these additional quantifications in our revised manuscript in the above-mentioned Figures and indicated the sample size (n) in their corresponding legends.

Minor Points

1) Please quantify P180 levels to verify knockdown efficiency (Figure 6).

R: We have performed Western blot analysis for P180 protein levels in control (pSuper) and P180 knockdown conditions in rat cells. We used a rat cell line (INS-1 cells) for this since it is difficult to achieve a high transfection efficiency in neurons, which would be required for this analysis. Both shRNAs against P180 show a significant decrease in P180 protein levels, thereby confirming that our knockdown is effective (Figure S5C).

2) In addition to microtubule stabilization, P180 has been reported to play a role in ribosome-independent mRNA localization to the ER (Cui et al., 2012). Please include discussion of other functions of P180 and how they may/may not interface with ER-lysosome contact dynamics and function.

R: We thank the reviewer for raising this interesting point. Indeed, P180 has been implicated in the regulation of mRNA localization and translation at the ER, although this has not been shown in neurons. We now have included a sentence about the possible relevance of these functions in our Discussion section.

3) The labeling in Figure 3C lower two panels should be corrected (i.e. change “SBP-RTN4A + KIF5A-FRB” to “SBP-RTN4A + KIF5A-Strep”).

R: We have corrected this mislabeling.

***** Reviewer #2 *****

How neuronal cells maintain an organized anterograde and retrograde flow of organelles within their axonal and dendritic compartments is a key but poorly understood problem. The manuscript by Ozkan et al proposes a role for tubular ER in mediating the transport of lysosomes, which are key mediators of cellular catabolism and metabolic signaling, within the axon. Building on recent work showing a key role for the ER in facilitating lysosomal fusion/fission, the authors provide evidence that physical contact between ER and lysosomes at the pre-axonal region promotes loading of a specialized kinesin onto lysosomes, resulting in their fission and movement down the axon. The final model proposes that the ER may thus act as a gatekeeper that regulates the number and size of lysosomes that are allowed into the axon. This is a very interesting and well executed manuscript that is supported by high quality biochemical and imaging data. A few suggestions for improvement are listed below.

R: We appreciate the reviewer for her/his positive comments and constructive feedback on our manuscript. We have performed new experiments based on this reviewer’s comments and we believe that the requested experiments further strengthen our conclusions.

1. In Fig. 1 and subsequent figures, the authors show that knocking down ER-organizing factors such as RTN4 and DP1 causes clear changes in the distribution of lysosomes and their morphology. Given the role of lysosomes in supporting mTORC1 signaling, the authors should check the status of canonical mTORC1 substrates (i.e. S6K, 4EBP1, TFE3) upon depletion of these factors.

R: We thank the reviewer for raising this interesting point. Lysosome stress has been shown to disrupt mTORC1 signaling, in which mTORC1 substrates such as TFE3 and TFE3 are translocated / sequestered into the nucleus (Martina et al., 2014; Bordi et al., 2016). Since we observe enlarged lysosomes in the soma after ER tubule disruption, it is possible that mTORC1 signaling is compromised. TFE3 has previously been shown to be sequestered in the nucleus where it can regulate expression of genes related to autophagy and lysosome biogenesis upon mTORC1 inactivation after nutrient depletion or lysosomal stress (Martina et al., Sci. Signal, 2014). We have tested an antibody against TFE3 in control neurons and upon RTN4 plus DP1 knockdown. From three independent experiments, TFE3 was found to be diffuse in cytosol in 141 transfected control neurons and 124 transfected neurons with shRNAs against RTN4 and DP1. We did not observe TFE3 translocation to the nucleus in any of the transfected cells. We have included representative images of the cytosolic TFE3 localization in both control and knockdown neurons in Figure S4D. We have also evaluated autophagy up-regulation related to this point, which is explained below in point 2.

2. Connected to the previous point, do these alterations of ER morphology and the resulting impairment of lysosomal positioning impact autophagy initiation, progression or termination? This could be ascertained using markers such as the green-red-LC3, as well as staining for Atg13 puncta.

R: As suggested by the reviewer, we have tested the GFP-LC3-RFP marker (Addgene #84573) which is often used in cell lines, where it labels autophagosomes in yellow and autolysosomes in red because GFP is cleaved upon fusion of autophagosomes and lysosomes. However, in our neurons we observed not only yellow or red-positive compartments, as reported, but also a strong GFP cytosolic signal that has made it impossible to evaluate and quantify autophagy with this system. We also expressed different concentrations of mCh-LC3 plasmid together with LAMP1-GFP to study autophagy, but in our hands mCh-LC3 distribution was very variable and remained mainly cytosolic in most of the cells. In addition, we tested a labelling kit (CytolID; EnzoLifeSciences) for the detection of endogenous autophagosomes, but the kit was not working in neurons in our hands. To be able to answer this question, we have performed endogenous p62 labeling in neurons expressing LAMP1-GFP together with a control pSuper vector or vectors containing shRNAs against RTN4 plus DP1. We quantified the total number of p62-positive puncta (autophagosomes and autolysosomes) and the number of p62-puncta colocalizing with LAMP1 (only auto-lysosomes) for both conditions. We did not observe any significant change in total number of p62 puncta nor an increase in p62-positive/LAMP1-positive puncta (Figure S4F-H). In addition, and related to point 1, we analyzed our previous quantitative proteomics data for control, RTNs and DP1 knockdown neurons (published in Farías et al., Neuron, 2019) to determine if autophagy-related proteins were increased upon ER tubule disruption as a consequence of lysosome-stress response. Protein levels of autophagy markers such as p62, LC3 and Atg13 were not affected. These results, together with the results mentioned in point 1, suggest that the enlarged lysosomes observed upon ER tubule disruption may not produce lysosome-stress-mediated autophagy up-regulation. We show these results in Figures S4D-H and we describe them in our revised manuscript.

3. Although the evidence supporting the requirement for P180 in lysosomal loading onto microtubules is strong, the exact relationship between P180 and kinesin-1 remains unclear. Recently, the multisubunit BORC complex was shown to promote kinesin loading onto lysosomes via the small GTPase Arl8 (PMID: 25898167). Does P180 interact with BORC or Arl8?

R: We agree with the reviewer that it would be interesting to further investigate the exact relationship between P180 and kinesin-1 and the possible interaction with lysosome adaptor proteins. This proposed experiment that would be needed to answer this question would require biochemical and/or proteomic approaches that require high transfection efficiency, which is difficult to achieve in primary neurons. Moreover, we believe this topic is beyond the scope of the current manuscript but could be very interesting as a follow-up study. We have added possible links in our discussion between the ER, P180 and BORC-Arl8-SKIP complex at the pre-axonal region.

4. Given the role of ER in mitochondrial homeostasis, and the feedback effects that mitochondrial status exerts on lysosomes (i.e. via AMPK) it would be good to test mitochondrial function upon depletion of the ER-organizing factors, including membrane potential and redox status.

R: We thank the reviewer for this comment. We have analyzed the distribution of mitochondria upon ER tubule disruption and ER tubule repositioning from the soma into the axon. These experiments have shown that distribution of mitochondria in neurons is not affected by RTN4/DP1 knockdown nor ER repositioning into the axon (Figures S2C, S2D, S2G and S2H). Although we agree it would be interesting to further investigate the effect of ER tubule disruption and repositioning on mitochondrial status, we believe it is beyond the scope of the current manuscript.

***** Reviewer #3 *****

In their manuscript titled “ER – lysosome contacts at a pre-axonal region regulate axonal lysosome availability”, the authors explore a connection between lysosome transport in the axon and ER tubules in the soma of cultured neurons. They broadly conclude that somatic ER tubules regulate lysosome size and axonal translocation by promoting lysosome homo-fission, and suggest a model wherein ER tubule – lysosome contacts at the somatic pre-axonal region promote kinesin-1-powered lysosome fission and subsequent axonal translocation. These claims are novel and would be of interest to cell- and neuro- biologists. Unfortunately, I did not find that the experimental data supports either of these broad conclusions.

R: We thank the reviewer for his/her assessment that our claims are novel and of wide interest to cell- and neurobiologists. As we will make clear in our response to this reviewer’s comments below, we do however believe our findings support our claims, and based on his/her comments, we have now performed additional experiments to further strengthen our proposed model.

Major comments

ER-shape regulates lysosome availability in the axon

1. The authors conclude from the experiments summarized in figure 1 that they show “ER tubules play a critical role in regulating lysosome translocation from the soma into the axon. “ This conclusion is not supported by the data. The authors do not show that the phenotype of RTN4 + DP1 KD is exclusively or even mainly a reduction in ER tubules. Yet, they draw a direct link between the KD and the morphology of the ER. While this may be one of the phenotypes supported by the literature, interfering with ER morphology would have a pleiotropic effect on the cell. In particular membrane proteins like LAMP1 and on the homeostasis of the entire endomembrane system, of which lysosomes are only one branch. How did the authors exclude an indirect effect that would explain the results?

R: We agree with the reviewer that it is necessary to verify the effect of RTN4 plus DP1 KD on ER tubules. Regarding change in ER shape, we have previously shown that ER morphology can be altered by manipulating the levels of ER-shaping proteins in neurons (Fariás et al., Neuron, 2019). More specifically, we have shown that RTN4/DP1 KD leads to a reduction in ER tubule membrane itself along the axon, which is the only ER shape present in the axon (Fariás et al., Neuron, 2019). To further support our claims that ER shape change upon RTN4/DP1 knockdown, we have performed two additional experiments. In the first experiment, we have studied ER morphology at nanoscale resolution using the newly developed Ten-Fold Robust Expansion (TReX) microscopy (Damstra et al., 2021; BioRxiv) in control and RTN4/DP1 knockdown neurons using Sec61β-GFP as a general ER marker. This revealed a clear change in ER morphology upon RTN4/DP1 knockdown with a reduction in tubular ER and increased appearance of large sheet-like structures at the soma (Figure S1 and Video S1). In the second experiment, we have performed live cell imaging analysis with a focus on the somatic pre-axonal region upon RTN4/DP1 KD. In control neurons, dynamic ER tubules labelled with the general ER marker Sec61β are localized to the pre-axonal and proximal axon regions, as shown with the AIS marker, TRIM46. Upon RTN4/DP1 KD, dynamic ER tubules were reduced at the pre-axonal and proximal axon regions and Sec61β was mainly localized in stable sheet-like structures at the soma. Together, this shows that RTN4/DP1 KD is clearly affecting ER morphology and supports our claims.

Regarding pleiotropic effect and homeostasis of the endomembrane system, it is indeed possible that interfering with ER morphology has a pleiotropic effect and/or change the homeostasis of the entire endomembrane system, which could indirectly explain the reduction in lysosome availability in the axon. In order to examine

this possibility, we have studied the effect of ER tubule disruption on cell viability, as well as in the distribution of different organelles such as mitochondria, Rab3-positive synaptic vesicles, Rab11-positive recycling endosomes. Our analysis revealed that knockdown of the ER tubule-shaping proteins RTN4 and DP1 did not alter cell viability nor distribution of mitochondria, Rab3 and Rab11 in axons as shown in new Figures S2B, 2C, 2E, 2F, 2G, 2I and 2J. We have also included in original submission staining and/or expression of early endosomes, recycling endosomes and autophagosome markers together with lysosomal markers in the soma. Beside lysosomal markers contained in enlarged compartments, other endosomal markers showed normal appearance, and they were not particularly enriched in the LAMP1+ or LAMTOR4+ enlarged compartments (original Figures S1C-E, now showed as Figures S4A-C). In addition to this, we have also studied possible lysosome-stress related autophagy up-regulation, but ER disruption did not cause any apparent increase in autophagy (new Figures S4D-H). We have added all these new data to the above-mentioned Figures, and they are described in our revised manuscript.

2. I am not convinced the polarity index calculation can be taken as a surrogate for lysosome distribution (figure 1). The mean fluorescence intensity would correlate with the average size of the lysosomes as well as their distribution. Consider a scenario where the amount of lysosomes in the dendrite is much larger but they are all dimmer than those of in the axon? How do the authors assure this is not the case? The authors should show the distribution of fluorescence intensities in axonal and dendritic lysosomes. In figure 3 the authors actually show phenotypes associated with lysosome size using the same shRNA.

R: We understand the reviewer's concerns regarding our polarity index calculations. However, we think it is highly unlikely that our findings can be explained by larger and dimmer lysosomes in dendrites. Indeed, we observe a phenotype in lysosome size after knockdown in the soma, but we do not observe these enlarged lysosomes in dendrites when analyzing higher magnification images as shown in a representative image in Figure 1A. As this reviewer suggests, we have included intensity profile plots for both dendrites and axons in new Figure S2A. We also show that the total number of lysosomes in a segment of the proximal axon are drastically reduced (Figure 1F). We have added the total number of lysosomes visualized during our 5-min live cell imaging in a region of proximal axon for all our manipulation systems (Figures 1F, 2F, 6B).

Somatic, but not axonal, ER tubules promote lysosome translocation into the axon

3. How do the authors exclude the possibility that when they induce axonal transport of ER tubules (Figure 2; "+ Strep-KIF5A") the phenotype they observe is not due to lack of space or lack of available motors for the lysosomes? Basically, why do they think it is anything beyond a simple traffic jam.

R: We appreciate this question from the reviewer. To exclude the possibility of a 'traffic jam', we have examined mitochondrial distribution upon somatic ER tubule translocation into axon. This did not reveal any alterations in mitochondrial transport into the axon. Since mitochondria are larger in size than lysosomes and both organelles use the same motor protein (kinesin-1) for their transport into the axon (van Spronsen et al., Neuron, 2013; Farías et al., PNAS, 2017), and lysosomes but not mitochondria are impaired in their axonal distribution, a lack of space and available motor is unlikely the case. We have provided representative images and quantification in new Figure S2D and S2H).

Local ER tubule disruption causes enlarged and less motile mature lysosomes in the soma

4. In Figure 3 – Even if the author’s assumption that RTN4 + DP1 KD leads to a reduction in tubules. Would not the reduction be everywhere and not just in the soma? Can the authors offer some explanation why the phenotype of the KD is similar to the somatic reduction alone? The images actually suggest the lysosomes are larger under shRNA conditions than under somatic reduction alone.

R: Regarding assumption that RTN4+DP1 KD leads to a reduction in ER tubules, we addressed this in point 1. As we have shown in our previous work (Farias et al., Neuron, 2019) and in Figure S1, Video S1 and S2, RTN4 + DP1 KD leads to a reduction in ER tubules in both the soma and the axon. In order to reveal if somatic or axonal ER tubules elicit the effect on lysosomal size and axonal distribution, we therefore manipulated the position of ER tubules using our heterodimerization system to reduce local axonal or somatic ER tubules (Figures 2 and 3C). This revealed that pulling ER tubules from the soma into the axon, but not from the axon into the soma, leads to reduced lysosome transport into the axon, and enlarged and less motile lysosomes retained in the soma (Figure 2 and 3, and new Videos S6), which suggests the effect of ER tubule disruption on lysosomes is likely caused by disrupting somatic, but not axonal ER tubules. The effect on lysosome size and distribution after reduction of all ER tubules in the RTN4 + DP1 KD condition is therefore likely mainly driven by somatic ER tubules, while axonal ER tubules play no or a minimal role in lysosome size and distribution.

The observation that lysosomes are possibly larger after RTN4 + DP1 KD than under somatic reduction alone can be well explained by the fact that we are knocking down RTN4 and DP1 for a period of 4 days, whilst somatic ER tubule reduction/repositioning using our heterodimerization system was done for 1 day. We have included additional data, quantification and description for lysosome number and size expressing our pulling system for 2 days in which somatic ER tubule repositioning causes similar phenotype than RTN4+DP1 KD, which is significantly increased compared to control (new Figures S3C-F and new Video S6).

The importance of somatic ER tubules is further supported by our analysis of the location of ER-lysosome contact sites using our split APEX system (Figure 5). This showed that contacts between the ER and lysosomes are mostly formed in the soma, particularly at the pre-axonal region (see also new Figure 5I-K). These results are consistent with previous findings showing ER-lysosome contacts are mainly formed in the cell body in neurons (Wu et al., PNAS, 2017). RTN4+DP1 KD causes a disruption of these contacts and results in enlarged lysosomes (Figure 5G). We have now added new data visualizing fission and translocation events occurring at ER – lysosome contacts in the somatic pre-axonal region in control cells, as well as shown impaired contact formation together with enlarged and less motile lysosomes in the same region upon RTN4+DP1 KD, by live cell imaging (new Figure 7, new Video S11). We have described the new data in our revised manuscript.

5. I did not understand how the authors conclude that on the one hand “ER tubule disruption did not affect lysosome activity “ and on the other hand that “mature lysosome population (LAMP1 / SirLyso positive) under ER tubule disruption was reduced to 33% “. Mature lysosomes and lysosome activity were defined by the same method (SirLyso). I would argue that lysosome activity is the cumulative activity of the lysosomes in the cell and not the individual activity of each lysosome. Even if I was to accept the later, the authors only show that some activity is measurable without truly quantifying SirLyso or MagicRed fluorescence intensity or showing that cathepsin translation and transcription are unaltered under these experimental conditions. The authors should offer some validation of their experimental model in these regards, preferably using standard biochemistry.

R: Indeed, we find that the population of mature lysosomes (LAMP1/ SirLyso positive) is reduced after RTN4 + DP1 KD to 33% compared to control neurons (Figure 3E). However, since the total population of lysosomes is also decreased, the relative proportion of active lysosomes is not decreased after ER tubule disruption (see

Figure 3F). This Figure now includes quantifications from an additional set of neurons to further strengthen these findings. The decreased population of lysosomes are larger in size and therefore the activity of each individual lysosome is likely increased after ER tubule disruption. That said, the reviewer is correct that these quantifications do not reflect the cumulative activity of lysosomes. We therefore quantified SirLyso (Cathepsin-D activity) fluorescence intensities in control and RTN4+DP1 KD neurons as suggested, and we did not observe any significant differences between control and RTN4+DP1 KD neurons. We have included the graph of this quantification in new Figure S3J. To provide further support that ER tubule disruption does not affect cathepsins, we analyzed our previous quantitative proteomics data for control, RTNs and DP1 knockdown neurons (published in Farías et al., Neuron, 2019). Protein levels for Cathepsin-B and -D were not reduced compared to control neurons (new Figure S3K)

6. Given that the authors disrupted ER morphology how can they be sure the LAMP1 signal still defines lysosomes? In particular, in light of the fact that LAMP1 and SirLyso don't fully overlap under RTN4 + DP1 KD. I cannot judge it for the somatic depletion (figure 3C) because the data is not shown. The authors casually state that "lysosome activity was often observed compartmentalized" but offer no evidence that these can be defined as lysosomes at all and not an artificial intermediate. CLEM data could have been potentially used to address this directly.

R: We appreciate this comment addressing a concern about whether lysosomal marker LAMP1 that we have used still defines the lysosomes. We have shown enlarged lysosomes not only with LAMP1 but also with LAMTOR4 marker (Figure 3B and current Figure S4C). As explained earlier, we also showed not enriched distribution of other endosomal markers in these enlarged structures, suggesting this is not form as an artificial intermediate endosomal compartment (Figures S4A-C). In addition, we showed CLEM experiment, which is a direct proof that LAMP1 signal still defines lysosomes. Correlation of LAMP1 signal in confocal imaging, as well as the SirLyso signal, corresponds to the lysosomes in the EM images. We have shown that activity is compartmentalized in LAMP1-positive lysosomes by CLEM and observed that degradative material inside the same compartment is delimited by a single membrane (Figure 3J-O). We have now included CD63 lysosomal marker together with SirLyso, in which similar results of enlarged lysosomes with compartmentalized SirLyso activity were observed (new Figure S3I, top panel). Related to the concern of no showing SirLyso activity after somatic ER tubule disruption, we have now included representative images of SirLyso and LAMP1 with RTN4-SBP+Strep-KIF5A expression, which shows SirLyso compartmentalized in enlarged LAMP1-positive structures (new Figure S3I, bottom panel). New data is included in the above-mentioned Figures and described in our revised manuscript.

7. I also have major concerns regarding the CLEM experiments. I did not understand why CLEM is used to measure the diameter of micron sized objects? Can the authors show what is the difference if the diameter is measured from the fluorescence data? The advantage being that FM data does not undergo shrinkage. The authors should also measure the diameter of lysosomes in the control. I did not understand where 400nm came from. I also did not find any statistics. How many times the experiment was repeated, how many cells were analyzed etc. The authors state that "The compartmentalized fluorescence SirLyso signal corresponded to the areas with intraluminal vesicles ". I wonder if the authors have enough resolution and correlation precision to support this statement. The correlation was done manually on resliced images so how can they be sure. Take for example lysosome number 2. I see no correspondence between the SirLyso and intraluminal vesicles.

R: We appreciate the concerns related to CLEM experiments. CLEM experiments were performed mainly to correlate localized fluorescence microscopy (FM) of LAMP1 and SirLyso with morphological characteristics of the organelle at high spatial resolution. This experiment is extremely complex to perform in primary neurons in culture and time-consuming, so it would be very difficult to quantify lysosome size from several neurons and from independent experiments, as we do for all our fluorescence microscopy data. We therefore removed the size measurement of lysosomes done by CLEM and we only provide the size analysis from fluorescence microscopy in our revised manuscript. We have now included FIB-SEM for control neuron that was under processing at the time of our initial submission (new Figure S3L) to show lysosome morphology in the control condition in an experiment that was performed at the same day as for the RTN4/DP1 KD neuron.

Regarding compartmentalization of SirLyso signal in LAMP-positive lysosomes, the resolution of EM presented in these images is 5nm isotopically, and the intraluminal vesicles in organelles are clearly discernible. Thanks to the enlarged size of the lysosomes in RTN4+DP1 KD cells, the correlation of these cells was a rather easy task. Our correlation precision in x,y is definitely below 1 μ m, which was required to overlay single lysosomes in these images. The overlay precision in z is ultimately limited by the z-resolution of FM around 500nm. Basically 100 EM slices represent a single confocal FM slice. Addressing this, we have used the FM maximum intensity projection and screened all the EM slices for each corresponding x,y correlation. Once the xy correlation is assured, each lysosome is axially traced back in z, and the exact z position (the top and the bottom of a lysosome) is defined within the 3D EM stack. Actually, lysosome 2 exactly demonstrates what is described above. SirLyso signal is concentrated on the upper half of the organelle in FM, whereas the lower half is devoid of SirLyso signal. When we check the xz plane corresponding to the upper half of the lysosome 2 (depicted with the yellow line in Figure 3L and shown in Figure 3M), we see the lighter lumen and the presence of intraluminal vesicles. On the contrary, when we check the xz plane corresponding to the lower SirLyso devoid half of the organelle (depicted with the blue line in Figure 3L and shown in Figure 3N), we see the darker lumen representing degraded material).

8. “Since RTN4/DP1 knockdown also reduced the total number of lysosomes (Figure 3E)”, how can the authors conclude anything about their distribution (figure 1) without normalizing the data on the overall number?

R: We understand the concerns regarding with normalization on overall number of lysosomes. However, the reviewer should take into account that neurons are highly polarized and complex cells with axons reaching up to 20 mm in length and with multiple branches in dendrites and axon. Therefore, it is nearly impossible to count the total number of lysosomes in one neuron manually, and an automatic count cannot be optimized due to the lack of imaging technology that can capture whole neurons with all its dendrites and its axon at enough resolution and sensitivity. The number of lysosomes provided in Figure 3E are the total number of lysosomes in the soma only. We think reviewer’s concerns with our conclusion of the effect of RTN4+DP1 knockdown on lysosome distribution may be based on a misunderstanding regarding our findings and our suggested model. With the hope of providing clarification, we now provide a graphical representation of our model (new Figure 7E), in which we suggest that firstly, contacts between ER tubules and lysosomes in the soma, mainly enriched at pre-axonal region (which we have shown with our APEX assay in our manuscript) are necessary for ER tubule-mediated lysosome fission. Secondly, P180, which is also enriched at the pre-axonal region, and drives kinesin-1 loading onto the lysosome that mediates the final fission step followed by lysosome translocation into the axon. In case of somatic ER tubule disruption, loss of ER-lysosome contacts leads to an impairment in lysosome fission, which causes lysosome enlargement at the soma and a subsequent impaired lysosome translocation

into the axon. We have included this model in Figure 7E, and we hope this clarifies our suggested model, which is based on original and new data added to our revised manuscript.

ER tubules regulate lysosome homo-fission

9. How can the conclusion from these results even remotely suggest anything about ER tubule – lysosome contacts. The authors should really be more precise throughout the manuscript and avoid jumping to conclusions.

R: We are slightly puzzled by this comment as we directly show that ER tubules and lysosomes form contacts at the somatic pre-axonal region using split APEX (Figure 5) and these contacts are decreased after ER tubule disruption (Figure 5G). These contacts are known to be important for lysosome fission and we show that ER tubule disruption affects lysosome size and homo-fission (Figure 3 and 4), lysosome distribution and transport into the axon (Figure 1 and 2). In addition, we now provide additional evidence of ER-lysosome contacts at the pre-axonal region and the effect of RTN4/DP1 and P180 KD on these contacts using live-cell imaging of the ER, lysosomes and the AIS and using the GB/RA reversible contact assay as explained in our response to point 8 of reviewer 1. These data are shown in new Figure 7 and Video S11. Together, we believe this provides sufficient evidence to support our model.

Minor comments

1. The paper is well written but can improved with less Jargon and by explaining the rationale of the experiments. Otherwise it may not be suitable to a broad audience.

Examples:

a. day-in-vitro 7 (DIV7) – would only be obvious to someone in the field.

b. “Quantification of the polarity index” why was it calculated, what was the rationale?

R: We thank the reviewer for these comments. However, ‘day-in-vitro’ (DIV) is a general and widely used term in many research papers in the cell and neurobiology field. To clarify this for a broader audience, we have included a more detailed explanation of this in the Methods section.

The quantification of the polarity index is a common method that is widely used to indicate the distribution of proteins and organelles in highly polarized and complex cells such as neurons (Kapitein et al., Current Biology, 2010; van Spronsen et al., Neuron, 2013; Karasmanis et al., Developmental Cell, 2017; Fariás et al., Neuron, 2019, Tortosa et al., Neuron, 2018; Pan et al., Cell Reports, 2019). In addition to the formula for this calculation that is currently in both the main text and methods section of the manuscript, we have clarified the rationale in Result and Method sections in our revised manuscript.

2. Unless citations are highly limited Fiji/ImageJ and all plugins used should be acknowledged by citing the associated publication.

R: We have now included a reference for Fiji/ImageJ in the Methods section of the revised manuscript.

3. In ShRNA experiments the details of the control are not provided. The standard control for ShRNA is a non-targeting shRNA sequence. Minimally, an empty vector control would be acceptable.

R: We apologize if this was not sufficiently clear in the submitted manuscript. All shRNAs were subcloned in a pSuper plasmid and we indeed used an empty pSuper plasmid as a control for all knockdown experiments. In the initial submitted manuscript we described this for each experiment in the Figure legends and we provided the sequences that were used to generate the shRNAs in Methods section. We now mention the use of an empty pSuper plasmid as a control in the Results section as well.

4. It should be clarified in the text that Figure 1 A and C and figure 3 A and B are the same experiment. The only difference is in what was quantified.

R: The reviewer is indeed correct that the transfections and immunostainings in Figure 1A, 1C and 3A and 3B are from the same conditions. However, the focus in Figure 1A and 1C is on the distribution of lysosomes in the entire neuron so confocal imaging was performed with lower magnification whilst the focus for figure 3A and 3B is the lysosome morphology in soma; therefore, the confocal imaging was performed with higher magnification by focusing on the soma and z stacks were performed to get better images. We believe this is already well explained in the Figure legends.

5. The authors may consider always having an ER tubule fluorescent reporter as a co-transfection marker for shRNA experiments.

R: We thank the reviewer for this suggestion and agree it is useful to do this when possible. However, limitations in the number of channels that can be used for confocal and live cell imaging prohibits us to do this for every experiment. For clarification, in primary cultures of neurons, the co-transfection efficiency of different plasmids is very high, so most of the labelled neurons also express our shRNAs. In the revised manuscript we have used the general ER marker Sec61-GFP in experiments shown in Figure S1, Video S1, Video S2 to reveal the change in ER shape upon RTN4/DP1 knockdown.

6. The statement "lysosomes of remarkable consistent size and shape" is imprecise and should be removed. The lysosomes in question are simply spherical and their size distribution is actually larger than the overall size of the WT lysosomes (400nm), so how can it be defined as consistent. Moreover, the number of measurements done seems extremely small and ignores the majority of LAMP1 fluorescent puncta.

R: The reviewer is correct and as we mentioned in our response to point 7, we have decided to remove lysosome size measurements based on CLEM data and description from the manuscript. We have also adapted the description of lysosome morphology, which is referring only to the cluster of enlarged lysosomes shown in CLEM image. In addition, as mentioned in our response to point 7, we have now also included EM images from a control neuron (Figure S3L).

7. The number of cells quantified and the number repeats should be stated for each experiment.

R: We have now more clearly stated the number of cells quantified in both the figure legends and methods in the revised manuscript.

8. Rab7 is a classical late endosome marker and not a lysosome marker, unless it co-localizes with other lysosomal markers like LAMP1. RAB7 overexpression has been recently shown to induce the formation of triple contact sites between ER late endosomes and mitochondria (see <https://doi.org/10.1073/pnas.1913509116> and <https://www.nature.com/articles/s41467-020-17451-7>).

We thank the reviewer for notifying us about these articles. We would like to clarify that the term 'lysosome' used in our manuscript, includes late endosomes and lysosomes (immature and mature lysosomes, respectively) as explained in the beginning of our result section. Regarding Rab7 expression, we use a Rab7 plasmid in our split APEX experiments (Figure 5) where we use endogenous LAMTOR4 to mark lysosomes. We do not believe our use of Rab7 overexpression is an issue in our split APEX experiments since it would not explain a specific enrichment of ER-lysosome contacts in the pre-axonal region, and its reduction upon RTN4/DP1 knockdown.

9. Can the authors not use CLEM to demonstrate the existence of contact sites at the pre-axonal region?

R: The reviewer is right that CLEM can be used to visualize ER-lysosome contact sites. However, as the reviewer may know, CLEM is not a high-throughput method, but a rather targeted and time-consuming method. Therefore, it is not suitable for quantification of ER-lysosome contacts. We have optimized and used two highly innovated and brand-new tools to demonstrate and monitor the ER-lysosome contact in neurons. We have shown that ER-lysosome contacts are mainly present in the soma and particularly enriched at a pre-axonal region with the use of the Split-APEX assay in our revised manuscript (Figure 5H-J). We also used the reversible GB/RA contact assay to monitor ER-lysosome contact in live neurons and observed ER-lysosome contacts at pre-axonal region (new Figure 7 and Video S11). Finally, we performed new live-cell imaging experiments using Sec61-GFP, SirLyso and TRIM46-BFP to visualize ER-lysosome contacts and their dynamics at the pre-axonal region (new Figure 7A and Video S11)

***** Reviewer #4 *****

In nonneuronal cells, it is known that ER-lysosome contact sites are required to enable lysosome homo-fission, and kinesin is involved in lysosome fission. ER structure is highly dynamic; it undergoes remodeling in the order of seconds. It is not clear how the local ER organization may control the LE/lysosome size and function. Here the authors show that in rat hippocampal neurons, knockdown of the ER tethering proteins VAPs do not produce disruption in ER lysosome contact, suggesting that redundancy may exist in membrane tethering events that involve ER. Instead, knockdown of proteins that control the ER shape, RTN4, and DPI, cause ER tubule disruption, causes disruption in ER-lysosome contact sites, and interferes with lysosomal homo-fission. They also show that in neurons, the ER-protein P180 binds microtubules to promote kinesin-1 dependent lysosome fission. This work emphasizes the importance of ER shape in controlling interactions between the ER membrane and the endo/lysosome membrane. The experiments are extensively and carefully performed. The work is novel and interesting.

R: We thank the reviewer for his/her positive assessment and helpful comments and suggestions on our manuscript.

1. RESULT- Please report efficiencies in individual KD experiments. KD VAPs do not produce disruption in ER lysosome contact. How was the contact sites monitored? Could this be due to a lack of efficiency in VAP KD?

R: We agree with the reviewer that it is important to validate knockdown efficiencies. To this end, we have previously validated the knockdown efficiency for rat shRNA-RTN4 and rat shRNA-DP1 in our lab using quantitative mass spectrometry (Farías et al., Neuron 2019; Supplemental Figure S3). Rat shRNA-CLIMP63 has been previously validated by the Ehlers lab (Cui-Wang et al., Cell 2012; Figures 6F-G and 7G) and the same sequence has been used in our previous paper Farías et al. Neuron, 2019. The shRNAs we used against VAPA and VAPB have also been previously validated (Teuling et al., Journal of Neuroscience, 2007) and used (Lindhout et al., EMBO Journal, 2019). The rat shRNA against protrudin used in this study has also been previously validated (Shirane and Nakayama, Science, 2006). In addition, we now provide Western blot analysis from three independent experiments for P180 protein levels in control (pSuper) and P180 knockdown conditions (shRNA1 and shRNA2) in rat cells. We used the rat INS-1 cell line for these experiments since it is very challenging to achieve a high transfection efficiency in neurons, which would be required for this analysis. Both shRNAs against P180 show a significant decrease in P180 protein levels, thereby confirming that our knockdown is effective (new Figure S5C). We describe now the validation of rat shRNAs in the Method section.

In our initial submission, we did not provide any data for the effect of VAP KD on ER-lysosome contact. We assume the reviewer is referring to our polarity index calculations for LAMP1 upon VAP KD (currently Figure S5A-B). However, in the revised manuscript we now provide data for the effect of both VAPA/B KD and VAPB overexpression on ER-lysosome contacts using our split APEX assay. These experiments show that VAP KD slightly reduce ER-lysosome contacts (Figure S6J and S6K) and VAPB overexpression increases ER-lysosome contacts (Figure S6L and S6M).

2. Please provide evidence that ER shape is changed after KDs of RTN4 and DPI.

R: We agree it is very important to provide evidence for a change in ER shape upon RTN4/DPI KD. We have previously shown that ER morphology can be altered by manipulating the levels of ER-shaping proteins in neurons (Farías et al., Neuron, 2019). More specifically, we have shown that RTN4/DPI KD leads to a reduction in ER tubule membrane itself along the axon, which is the only ER shape present in the axon (Farías et al., 2019). In the revised manuscript, we have added two new experiments that provide proof of ER reorganization upon RTN4/DPI KD. First, we have studied ER morphology at nanoscale resolution using the newly developed Ten-Fold Robust Expansion (TReX) microscopy (Damstra et al., 2021; BioRxiv) in control and RTN4/DPI knockdown neurons using Sec61 β -GFP as a general ER marker. This revealed a clear change in ER morphology upon RTN4/DPI knockdown with a reduction in tubular ER and increased appearance of large sheet-like structures at the soma (Figure S1 and Video S1). In addition, we have performed live cell imaging analysis with a focus on the somatic pre-axonal region upon RTN4/DPI KD. Dynamic ER tubules labelled with the general ER marker Sec61 β are localized to the pre-axonal region that is preceding the AIS marker, TRIM46 in our control cells. Upon RTN4/DPI KD, dynamic ER tubules were impaired at the pre-axonal region and Sec61 β was mainly localized in sheet-like structures at the soma. Together, this shows that RTN4/DPI KD is clearly affecting ER shape.

3. ABSTRACT. I suggest that the words "KD proteins that control the ER shape, RTN4 and DPI, causes ER tubule disruption and causes disruption in ER-lysosome contact" be included in the abstract.

R: We appreciate the reviewers' suggestion. However, we believe our current abstract accurately describes our most important findings and we cannot add the suggested sentence to the abstract due to word limitations. In addition, we hope the reviewer agrees this is now sufficiently clear in the main text of our revised manuscript.

4. INTRODUCTION-It needs to be revised; sentences that describe the results should be deleted.

R: We believe that a small summary of the results in the last paragraph of the introduction is common practice in many research papers/journals and we believe this provides structure to the introduction and makes it easier to read.

REVIEWERS' COMMENTS

Reviewer #1 (Remarks to the Author):

I appreciate that the authors have attempted to address the majority of my comments. I still think that the manuscript does not represent a major advance in the field, but the experiments are now better controlled and explained.

Reviewer #2 (Remarks to the Author):

The authors have addressed my previous comments satisfactorily, I have no further concerns and support publication of this interesting manuscript.

Reviewer #3 (Remarks to the Author):

I acknowledge that the authors have added a lot of information to address my original concerns, and that the differences in our views appear to be too great to ever be fully addressed. Therefore, I think the manuscript is suitable for publication with minor comments.

Minor comments:

- The authors consider Late endosome (LE) and lysosomes (LY) the same, although these are in fact two different organelles that differ in their proteome, function, structure and appearance/density observed in EM micrographs. The authors should at the least discuss the literature suggesting that ER-LE and ER-LY contact sites are structurally and functionally distinct.
- As the authors claim involvement of undefined membrane contact sites/connections between the ER and LE/LY in LE/LY fusion, they should also discuss how they suggest the ER is controlling the HOPS complex/tethering proteins mediating LE/LY fusion.
- I believe the use of redundancy on p. 17 is inaccurate. Different tethers for membrane contact sites (MCS) between the ER and LE/LY have been suggested in the literature but no redundancy between these tethers in terms of their function or the conditions at which they form/dissociate.
- I was unable to follow why the authors think that "Identifying neuron-specific tethering proteins involved in the formation and maintenance of ER – lysosome contact sites will be an important future research goal." when "Protrudin-Rab7" is their prototype tether for all ER-LY MCS throughout the paper.
- The authors should note that VAPA/B are not "the major tethering proteins" in the case of the Protrudin-mediated MCS. As shown before (Raiborg 2015 and Elbaz-Alon 2020), Protrudin is an integral ER membrane protein that binds PI(3,5)P2 on LEs directly through a FYVE domain. Another tether (PDZD8) is also an integral ER protein and binds Rab7 directly to form An ER/LE MCS. On the other hand, VAPA/B are the main tether component for MCS with other organelles but this is not discussed anywhere as far as I could find KD/KO of VAPA/B could also have a pleiotropic effect on the system.

Reviewer #4 (Remarks to the Author):

The manuscript has been carefully revised.

Response to reviewers

REVIEWERS' COMMENTS

Reviewer #1 (Remarks to the Author):

I appreciate that the authors have attempted to address the majority of my comments. I still think that the manuscript does not represent a major advance in the field, but the experiments are now better controlled and explained.

R: We thank the reviewer for his/her time to review our revised manuscript and for the appreciation of the improvements made during revisions.

The comment about that our findings do not represent a major advance was also mentioned during our first revision in the main comment: *“Considering the emerging role inter-organelle contacts in neuronal function, this study is interesting, and the experiments are logical and well-performed. However, the finding that ER-lysosome contacts regulate lysosomal fission and axonal lysosomal availability is a modest advance considering previous findings from this group and others (Farías et al., 2017; Farías et al., 2019; Rowland et al., 2014; Allison et al., 2017)..”*. This comment was followed by specific comments/experiments suggested by this reviewer, which as the reviewer indicates now, we then attempted to address with further experiments and explanation.

We would like to explain why we believe this work is a major advance, considering previous evidence (such as the publications mentioned by this reviewer). These previous findings are already introduced and extensively discussed in our manuscript.

First of all, we would like to highlight that neurons are highly polarized cells with distinct morphological and functional compartments, in which local organelle distribution ensures a local response required for neuronal development and neuronal function. Thus, elucidating the mechanisms involved in organelle positioning in neurons is important for neuronal physiology. This knowledge could help to understand why the organization, distribution and function of organelles are impaired in some neurodegenerative diseases. This is described in the Introduction and discussion.

Regarding the referred publications; in Farías et al., 2017 and Farías et al., 2019, Kinesin-1 was identified as the motor responsible for the translocation of lysosomes and ER tubules into the axon, respectively. Farías et al., 2017 and Farías et al., 2019 also demonstrated the role of local axonal distribution of lysosomes and ER tubules in axonal development, respectively. To our understanding, this current work is the first evidence indicating a regulatory role of ER morphology on local lysosome availability in neurons, highlighting the importance of inter-organelle communication in regulating polarized organelle distribution. These findings have been discussed and contrasted in our manuscript.

Lysosome fission and transport have been reported in contact with the ER, spread along the entire cytoplasm in non-neuronal cells (Rowlands et al., 2014; Raiborg et al., 2015 and other

studies). We now have better highlighted these previous findings in our introduction. However, lysosome fission and translocation events at contact sites have not been previously connected. This is clearly stated in our manuscript. Our evidence indicates that in highly polarized cells, these contacts are spatially confined to the soma and enriched at a pre-axonal region, a region in which lysosomes undergoing fission and subsequently translocating into the axon were observed. Impaired ER-lysosome contact caused enlarged lysosomes unable to undergo fission and translocate into the axon.

Allison et al., 2017 also found enlarged lysosomes under mutations or KO of some ER-associated proteins that are linked to hereditary spastic paraplegia such as spastin mutations and REEP1 KO in neurons. Allison et al., 2017 propose that these ER-associated proteins affect lysosome function indirectly, in which the trafficking of cathepsins between Golgi-endosome is disrupted in non-neuronal cells. Dynamics of lysosomes in neurons was not evaluated in this study. We have found that ER tubule disruption mainly affects lysosome size (fission) and translocation, but not lysosome cathepsin levels or activity. This evidence is described and contrasted with our findings in the manuscript.

We have better highlighted these previous studies and contrasted them with the evidence in our revised manuscript. We thereby believe that we now provide a good overview of what is already known from literature and describe the advances made by the findings in our study in a measured tone.

Reviewer #2 (Remarks to the Author):

The authors have addressed my previous comments satisfactorily, I have no further concerns and support publication of this interesting manuscript.

R: We are happy to hear that we have addressed all previous concerns satisfactorily and thank the reviewer for supporting the publication of our manuscript.

Reviewer #3 (Remarks to the Author):

I acknowledge that the authors have added a lot of information to address my original concerns, and that the differences in our views appear to be too great to ever be fully addressed. Therefore, I think the manuscript is suitable for publication with minor comments.

R: We thank the reviewer for his/her comments and appreciation of the improvements made in the revised manuscript. We have further revised the manuscript based on this reviewer's minor comments as outlined below.

Minor comments:

- The authors consider Late endosome (LE) and lysosomes (LY) the same, although these are in fact two different organelles that differ in their proteome, function, structure and appearance/density observed in EM micrographs. The authors should at the least discuss the literature suggesting that ER-LE and ER-LY contact sites are structurally and functionally distinct.

R: We completely agree with the reviewer's comment. LE and lysosomes are two different organelles. However, considering their dynamic conversion/ gradient of maturation, many of the markers often used to identify these organelles (such as LAMP1, Rab7, LAMTOR4) are present in both late endosomes and lysosomes. Because of this, many publications refer to them as LE/LY, LE or LY, or, like us, just call them lysosomes after explaining they correspond to late endosomes or lysosomes (Pu et al., 2016; Saffi & Botelho, 2019). To classify them properly, we should assess LAMP1 and Rab7 distribution together with cathepsin activity (e.g., SirLyso, MagicRed), and correlate this with EM. Unfortunately, we cannot perform all our experiments with different endo-lysosomal markers and CLEM as we did for characterizing enlarged lysosomes in the soma. Although we have studied different markers for LE or LY, we cannot distinguish between ER-LE and ER-LY contacts. It has recently been shown that some tethering proteins such as PDZD8, also mentioned in one of the reviewer's comment, mediates Rab7-dependent interaction of the ER with LE and LY (Guillen-Samander, et al., PNAS, 2019).

We visualized LE/LY fission events associated to ER tubules in a pre-axonal region using LAMP1 marker and SirLyso probe, with both markers we observed LE/LY undergoing fission and translocation into the axon. We have added a sentence to our Discussion explaining our terminology, and our limitation to classify them as LE or LY. It remains unclear from our study whether specifically ER-LE and/or ER-LY contact sites contribute to LE or LY translocation into the axon.

- As the authors claim involvement of undefined membrane contact sites/connections between the ER and LE/LY in LE/LY fusion, they should also discuss how they suggest the ER is controlling the HOPS complex/tethering proteins mediating LE/LY fusion.

R: In light of word limitations for this manuscript, we have decided to remove a discussion of the possible connection of the ER with LE/LY fusion, which is not one of the main messages of our findings.

- I believe the use of redundancy on p. 17 is inaccurate. Different tethers for membrane contact sites (MCS) between the ER and LE/LY have been suggested in the literature but no redundancy between these tethers in terms of their function or the conditions at which they form/dissociate.

R: We agree with the reviewer and have now changed this sentence in Discussion.

- I was unable to follow why the authors think that “Identifying neuron-specific tethering proteins involved in the formation and maintenance of ER – lysosome contact sites will be an important future research goal.” when “Protrudin-Rab7” is their prototype tether for all ER-LY MCS throughout the paper.

R: We have removed this sentence to add other suggested points in our Discussion. As the reviewer mentions, we describe that protrudin is enriched in ER – LE/LY contact sites. However, protrudin knockdown does not disrupt axonal distribution of LE/LY, suggesting compensatory mechanisms could account for the transport of LE/LY into the axon under these conditions.

- The authors should note that VAPA/B are not “the major tethering proteins” in the case of the Protrudin-mediated MCS. As shown before (Raiborg 2015 and Elbaz-Alon 2020), Protrudin is an integral ER membrane protein that binds PI(3,5)P2 on LEs directly through a FYVE domain. Another tether (PDZD8) is also an integral ER protein and binds Rab7 directly to form An ER/LE MCS. On the other hand, VAPA/B are the main tether component for MCS with other organelles but this is not discussed anywhere as far as I could find KD/KO of VAPA/B could also have a pleiotropic effect on the system.

R: In the discussion we previously stated, *‘Several ER – organelle tethering proteins have been identified at contact sites, with the ER protein VAP playing a main role in ER tethering to multiple organelles as well as the plasma membrane (Wu et al., 2018)’*.

We now state: *‘Several ER – organelle tethering proteins have been identified at contact sites, with the ER protein VAP playing a broader role in ER tethering to multiple organelles as well as the plasma membrane (Wu et al., 2018)’*. In addition, we added a sentence discussing the possible pleiotropic effect that VAPA/B KD could have on the system.

Finally, we now also discussed the role of protrudin in more detail.

Reviewer #4 (Remarks to the Author):

The manuscript has been carefully revised.

R: We appreciate this comment from the reviewer and would like to thank the reviewer for his/her time to review our manuscript.